# Zika virus noncoding RNA suppresses apoptosis and is required for virus transmission by mosquitoes

Andrii Slonchak[1], Leon E. Hugo[2], Morgan E. Freney[1], Sonja Hall-Mendelin[3], Alberto A. Amarilla[1], Francisco J. Torres[1], Yin Xiang Setoh[1], Nias Y. G. Peng[1], Julian D. J. Sng[1], Roy A. Hall[1], Andrew F. van den Hurk[3], Gregor J. Devine[2] & Alexander A. Khromykh[1]✉

Flaviviruses, including Zika virus (ZIKV), utilise host mRNA degradation machinery to produce subgenomic flaviviral RNA (sfRNA). In mammalian hosts, this noncoding RNA facilitates replication and pathogenesis of flaviviruses by inhibiting IFN-signalling, whereas the function of sfRNA in mosquitoes remains largely elusive. Herein, we conduct a series of in vitro and in vivo experiments to define the role of ZIKV sfRNA in infected *Aedes aegypti* employing viruses deficient in production of sfRNA. We show that sfRNA-deficient viruses have reduced ability to disseminate and reach saliva, thus implicating the role for sfRNA in productive infection and transmission. We also demonstrate that production of sfRNA alters the expression of mosquito genes related to cell death pathways, and prevents apoptosis in mosquito tissues. Inhibition of apoptosis restored replication and transmission of sfRNA-deficient mutants. Hence, we propose anti-apoptotic activity of sfRNA as the mechanism defining its role in ZIKV transmission.

[1] The University of Queensland, Brisbane, QLD 4072, Australia. [2] QIMR Berghofer Medical Research Institute, Brisbane, QLD 4006, Australia. [3] Public Health Virology, Forensic and Scientific Services, Department of Health, Queensland Government, Brisbane, QLD 4108, Australia. ✉email: a.khromykh@uq.edu.au

Rapid growth of the human population accompanied with urbanization, rapid air travel and climate change is creating an environment conducive to the emergence of previously uncommon arboviruses, such as Zika virus (ZIKV)[1]. ZIKV, a member of the *Flavivirus* genus in the *Flaviviridae* family, is primarily transmitted to humans by *Aedes aegypti* mosquitoes[2]. It poses a substantial public health concern due to the congenital abnormalities associated with ZIKV infection during pregnancy[3]. Transmission of ZIKV to humans via mosquito bite requires ingestion of infected blood by mosquitoes, followed by initial viral replication in midgut, dissemination of the virus through the mosquito body, infection of salivary glands and secretion of infectious particles into saliva[4]. Therefore, it is important to understand host factors and immune mechanisms in mosquitoes that affect this progression and ultimately whether the virus can be transmitted.

Flaviviruses utilise multiple cellular processes to enable their replication[5]. In particular, we previously discovered that flaviviruses exploit the cellular mRNA decay pathway and utilise the host 5′–3′ exoribonuclease XRN-1 to produce flaviviral subgenomic RNAs (sfRNAs)[6]. While digesting genomic RNA of flaviviruses, XRN-1 stalls at the structured XRN-1-resistant RNA elements (xrRNAs) in the 3′ untranslated region (3′UTRs), which results in generation and accumulation of incompletely degraded viral RNA[6–10]. ZIKV contains two experimentally validated xrRNAs (xrRNA1 and xrRNA2) formed by stem loops SLI and SLII and an additional putative xrRNA3 formed by a dumbbell element, DB1 (Fig. 1a). It generates two sfRNA species—the predominant longer isoform sfRNA1, which is produced by

stalling of XRN-1 at the xrRNA1 and less abundant shorter sfRNA2, which is generated due to XRN-1 slipping through the xrRNA1 and stalling at the xrRNA2 located ~100 nts downstream[11].

The ability to produce sfRNA is highly conserved within the *Flavivirus* genus and has been reported for mosquito-borne, tick-borne, insect-specific, and flaviviruses without known vectors[12,13]. This implies that sfRNA should possess an important function in both arthropod and vertebrate hosts. Although previous studies suggested possible inhibitory effects of sfRNA on the RNAi response[14,15] and the Toll pathway[16], evidence to support these mechanisms are rather inconsistent between different studies[17] and the exact role of sfRNA in arthropods is still unclear. To gain a better understanding of the molecular processes targeted by sfRNA in mosquitoes, we designed sfRNA-deficient ZIKV mutants, assessed their replication in mosquito cells and in vivo and conducted transcriptome-wide gene expression profiling of infected mosquitoes. We show that sfRNA facilitates productive ZIKV infection in mosquitoes and is essential for viral transmission. We also demonstrate that sfRNA inhibits apoptosis in the infected mosquitoes by altering expression of the genes that control cell death and survival.

## Results

**Deficiency in both sfRNAs compromises ZIKV viability.** To elucidate the functions of ZIKV sfRNA, we first generated mutant viruses deficient in production of sfRNA isoforms by introducing point mutations that abolish XRN-1 resistance[11] into each or

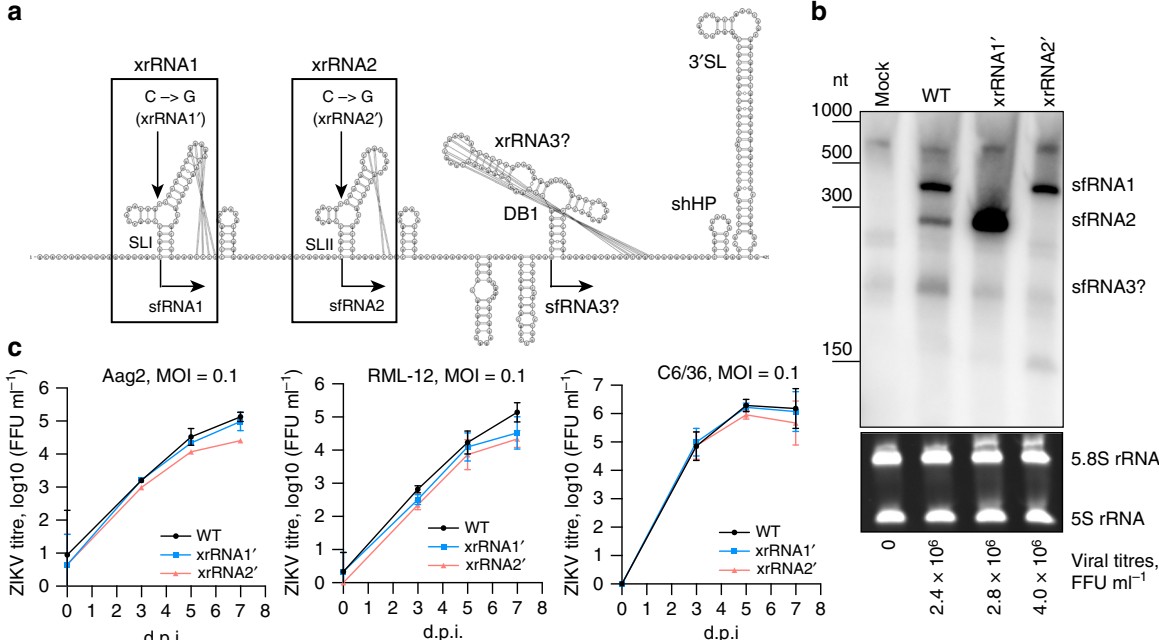

**Fig. 1 Analysis of sfRNA-deficient ZIKV mutants in cultured mosquito cells. a** Model of ZIKV 3′UTR secondary structure and location of mutations that impair sfRNA production. The model was created using the *mfold* algorithm, refined based on the crystal structure of xrRNA1′ and visualised with VIENNA RNA software. Mutation in xrRNA1 was described previously and is based on RNA crystal structure[11], mutation in xrRNA2 was designed based on homology between xrRNAs. SL, stem loop; DB, dumb bell; shHP, small hairpin; xrRNA, XRN-1-resistant RNA. **b** Production of sfRNAs by WT and mutant viruses in C6/36 cells. Cells were infected at MOI = 1, RNA was isolated at 3 dpi and used for Northern blot hybridization with radioactively labelled DNA oligo complementary to the viral 3′UTR. Bottom panel shows Et-Br-stained ribosomal RNA as a loading control. Viral titres shown below the panels were determined in culture fluids of the infected cells prior to RNA extraction from cells. **c** Growth kinetics of WT and sfRNA-deficient viruses in RNAi-deficient (C6/36) and RNAi-competent (RML-12 and Aag2) mosquito cell lines. Cells were infected at MOI = 0.1 and viral titres in culture fluids were determined at indicated time points. Titres in (**b**) and (**c**) were determined using IPA on Vero cells. Values in (**c**) represent the means from three independent experiments ± SD. Statistical comparison between mutants and WT virus was performed using two-way ANOVA with Geisser-Greenhouse correction for multiple comparisons. Image in (**b**) is a representative blot of two independent experiments that produced similar results.

both xrRNAs (xrRNA1 and xrRNA2) of the African MR766 strain of ZIKV (Fig. 1a). In order to produce these mutant viruses, infectious cDNA was assembled using the circular polymerase extension reaction (CPER)[18] (Supplementary Fig. 1A,B) and transfected into Vero cells. At 10 days post transfection (dpt), the presence of WT and recombinant ZIKV with mutations in either xrRNA1 (xrRNA1′) or xrRNA2 (xrRNA2′) in culture fluid of transfected cells was detected by RT-PCR for viral RNA (Supplementary Fig. 1B). Mutations in the corresponding xrRNAs of these viruses were confirmed by Sanger sequencing (Supplementary Fig. 1C) and their deficiency in generation of corresponding sfRNAs in infected mosquito cells was demonstrated by Northern hybridization (Fig. 1b). Therefore, we have shown that production of ZIKV sfRNA2 can be impaired by a point mutation in xrRNA2 similar to what was previously described for the generation of the sfRNA1-deficient mutant[11].

Although ZIKV mutants with single mutations in xrRNA1 or xrRNA2 were successfully generated and showed no attenuation in Vero cells (Supplementary Fig. 1D), the double mutant xrRNA1′2′ virus was not recovered and viral RNA was not detected in culture fluids of Vero cells transfected with the infectious cDNA (Supplementary Fig. 1B). This indicates that the inability to produce both sfRNAs completely attenuated the virus in mammalian cells. To account for potential host-specificity of the restriction imposed by the lack of both sfRNAs we also attempted to recover the xrRNA1′2′ mutant virus in mosquito cells. To increase the likelihood of virus recovery we used an RNAi-deficient mosquito cell line C6/36[19]. In this experiment, an insect OpiE2 promoter-based infectious cDNA[20] with xrRNA1′2′ mutations was generated by CPER and transfected into C6/36 cells. Starting from 7 days post transfection the viral RNA could be detected in the culture fluid of transfected cells by RT-PCR, however, sequencing revealed that the mutation in xrRNA2 had undergone a reversion to the WT sequence (Supplementary Fig. 1E). This experiment was repeated twice, and reversion was observed in both experiments.

Therefore, the virus needed to produce at least one sfRNA isoform in order to be capable of productive infection in either mammalian or mosquito cells. The occurrence of these reversion events suggests that although some replication of viral RNA was possible at an early stage of infection, it did not produce infectious viral particles given that the mutated genotype could not be detected in the culture fluid. It also indicates that the initial mutant phenotype was subjected to very strong negative selective pressure, which favoured the revertant over the mutant.

**ZIKV tolerates loss of individual sfRNAs in mosquito cells.** Given that sfRNA was previously reported to act in insect cells via inhibition of the RNAi pathway[14], we initially expected that viability of the double mutant deficient in production of sfRNA1 and sfRNA2 would not be affected in the RNAi-deficient mosquito cell line C6/36. However, the fact that only xrRNA1′ mutation was retained in C6/36 cells transfected with the xrRNA1′2′ cDNA indicated that certain antiviral mechanism(s) were acting in these cells against the sfRNA-deficient ZIKV but were evaded by the virus when at least one sfRNA was produced. To further assess the role of individual sfRNAs in ZIKV replication in mosquito cells and to investigate if the potential RNAi-suppressor activity of sfRNA may benefit viral replication, growth kinetics of WT ZIKV and single xrRNA mutants was examined in RNAi-competent (Aag2 and RML-12) and RNAi-deficient (C6/36) mosquito cells. The results demonstrated that although there was a trend towards attenuation in replication for xrRNA2′ mutant virus, the differences in replication between either of the mutants and WT ZIKV were not statistically significant for any of

the selected cell lines (Fig. 1c). The lack of differences between replication of WT virus and sfRNA mutants in either RNAi-deficient or RNAi-competent cell lines indicates that the RNAi-suppressor activity is unlikely to be a biologically relevant mechanism that determines the function of sfRNA in infection.

**sfRNA is required for ZIKV transmission by *Ae. aegypti*.** Deficiency in production of sfRNA was previously shown to be associated with reduced transmission of WNV by *Culex spp.* mosquitoes[17]. To investigate the role of sfRNA in mosquito transmission of ZIKV, female *Ae. aegypti* mosquitoes were exposed to WT, xrRNA1′ and xrRNA2′ ZIKV by blood feeding or intrathoracic (i.t.) injection (Supplementary Fig. 2A). Viral titres were then determined in bodies, legs and wings, and saliva of mosquitoes exposed to infectious blood meals at 7 and 14 days post infection (dpi) to examine viral replication, dissemination and transmission. At 7 dpi, virus was detected in the bodies of mosquitoes from all three groups with the infection rate for WT, xrRNA1′ and xrRNA2′ viruses being 20%, 23% and 10%, respectively (Fig. 2a). Notably, viral titres in the bodies of mosquitoes infected with mutant viruses were significantly lower than in the WT-infected group (Fig. 2b). At this time point ZIKV infection was primarily confined to the midgut and virus dissemination to legs and wings was not evident (Supplementary Fig. 3A,C). At 14 days after blood feeding, virus was detected in the bodies of 46% mosquitoes exposed to WT ZIKV (Fig. 2c). Infection rates in the groups exposed to xrRNA1′ and xrRNA2′ mutants were significantly ($P_{xrRNA1'} = 0.025$, $P_{xrRNA2'} = 0.0002$) lower compared with WT virus (23% and 20% body infection rates, respectively; Fig. 2c). In addition to the reduced infection rates, viral dissemination and transmission rates (presence of virus in saliva) in mosquitoes that fed on blood containing the sfRNA-deficient mutants was also significantly lower than in the WT virus-infected group (Fig. 2c, Supplementary Fig. 2B). Viral loads in the bodies of infected mosquitoes were highly variable within each group and no significant differences were identified between titres of WT and mutant viruses (Fig. 2d). Titres in legs and wings were more consistent within groups and were significantly lower for the xrRNA1′ mutant group compared with the WT virus group (Fig. 2d). Viral titres in saliva from mosquitoes infected with the xrRNA1′ mutant were also significantly reduced compared with the WT virus-infected group, and only one mosquito infected with xrRNA2′ mutant was virus-positive (Fig. 2d).

Although the xrRNA2′ mutant had a reduced ability to establish infection in the midgut, viral titres in legs and wings of mosquitoes inoculated with this virus were similar to those of WT ZIKV (Fig. 2d), which suggested that the xrRNA2′ mutation might have not been retained. Indeed, deep sequencing of ZIKV 3′UTR amplicons generated from viral RNA isolated from five individual ZIKV-positive xrRNA2′-infected mosquitoes confirmed that the C to G mutation in xrRNA2 had almost completely disappeared in all mosquitoes tested (G frequency in viral populations 1–8%). Instead, either the wild-type variant, C (42–62%), or A (31–54%) were found to be present in the corresponding position of viral genome (Supplementary Fig. 2C). In contrast, in the group exposed to the xrRNA1′ infection, all five analysed mosquitoes contained viruses that retained C to G mutation in xrRNA1. Four samples had G frequency of 90–93% and one mosquito contained a population of ~60% of mutant and 40% of WT viruses (Supplementary Fig. 2C). The expected sequencing accuracy was 94%[21]. Therefore, from the experiment with blood feeding, we can conclude that sfRNA1 facilitates productive infection and dissemination, and is required for efficient virus transmission. Although the effect of sfRNA2 on

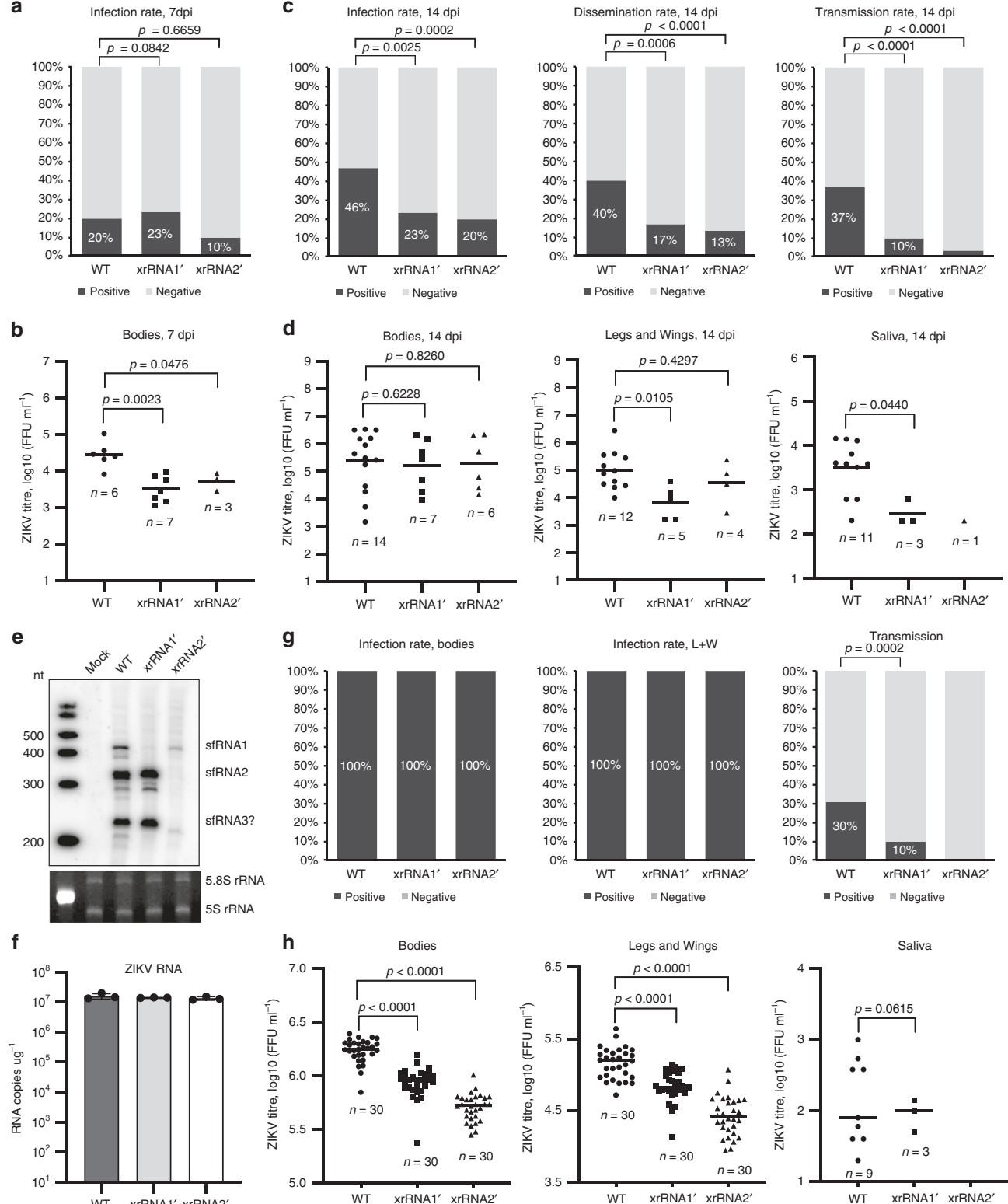

viral transmission could not be assessed in this experiment due to mutation loss, our data indicates that the xrRNA2′ mutation was detrimental for ZIKV infection in midguts as it was subjected to strong negative selection.

After inoculation via i.t. injection (10 dpi), xrRNA2′ virus retained the mutation (100% of viral population) in all examined mosquitoes (Supplementary Fig. 2D). In addition, Northern blot hybridisation of RNA extracted from i.t. injected mosquitoes

demonstrated that xrRNA1′ and xrRNA2′ mutants were deficient in production of sfRNA1 or sfRNA2, respectively, and therefore also had expected phenotypes (Fig. 2e). Northern hybridization analysis further revealed the production of an additional 3′UTR-derived RNA of ~200 nt in size in mosquitoes infected with WT virus and the xrRNA1′ mutant (Fig. 2e). A similar 200 nt band was also evident on the blot of the RNA from C6/36 cells, although it was less abundant (Fig. 1b). The size of this RNA

**Fig. 2 sfRNA facilitates replication and transmission of ZIKV in vivo.** *Ae. aegypti* mosquitoes were exposed to an infectious blood meal containing $10^8$ FFU/ml of each virus (≈1:5 mixture of virus stock and defibrinated sheep blood). At 7 days post infection (dpi), the percentage of infected mosquitoes (**a**) and viral titres in the bodies (**b**) were determined to serve as indicators of the infection rate and of initial viral replication. At 14 dpi (**c**), the infection, dissemination and transmission rates were determined as percentage of ZIKV-positive bodies, legs and wings, and saliva samples, respectively. ZIKV titres at 14 dpi (**d**) indicate viral replication efficiency and viral loads in saliva. **e** Production of ZIKV sfRNAs in infected mosquitoes. RNA was isolated from the pools of 10 mosquitoes at 10 days after i.t. injection and used for Northern blot hybridization with radioactively labelled DNA oligo complementary to the viral 3′UTR. The bottom panel shows a polyacrylamide gel with Et-Br-stained ribosomal RNA as loading control. Figure shows representative images of two independent experiments that produced similar results. **f** ZIKV genomic RNA levels in RNA samples used for Northern blot in (**e**) values are the means of three technical replicates +/− standard errors of the means. Viral RNA abundance was determined using qRT-PCR and the standard curve quantification method. Values are the means ± SEM of three technical replicates. **g** Infection and transmission rates in mosquitoes infected by i.t. injection. **h** ZIKV titres in mosquitoes inoculated by i.t. injection. Mosquitoes in (**e–h**) were injected with 200 nl of the inoculum containing $10^4$ FFU/ml of each virus. Viral titres were determined by IPA on C6/36 cells (**a–d**) or Vero 76 cells (**g**, **h**). Sample sizes (*n*) for all statistical tests indicated in the panels refer to biologically independent mosquitoes. Panels (**b**, **d**, **h**) show individual and median (horizontal line) values. Statistical differences were determined using chi-squared (**a**, **c**, **g**) or Mann–Whitney U tests (**b**, **d**, **h**), all *P*-values are two-sided, no multiple comparisons were performed in each test.

matched the approximate position of the putative sfRNA3 start site, at the beginning of DB1 (Fig. 1a) and therefore could represent a third sfRNA produced by stalling of XRN-1 at this DB1 structure. Production of a third sfRNA has not been previously reported from ZIKV-infected mammalian cells[11,22], which indicates that in mosquitoes XRN-1 may slip through the upstream xrRNAs more easily. We define this RNA molecule as sfRNA3. In addition to the expected deficiency in production of sfRNA2, xrRNA2′ virus replicating in vivo also had reduced production of sfRNA1 (Fig. 2e). Given that mosquitoes in all three groups contained similar levels of viral RNA (Fig. 2f), this may indicate the existence of interactions between xrRNA2 and xrRNA1, similar to those previously suggested for DENV[23].

At 10 days after i.t. injection, infection was established in bodies and legs and wings of 100% of mosquitoes irrespective of the virus (Fig. 2g). However, the titres of xrRNA1′ and xrRNA2′ mutants in mosquito bodies were significantly lower than in mosquitoes infected with WT ZIKV (Fig. 2h). Consistent with the observations from infection via blood feeding, the titres of both sfRNA-deficient mutants in legs and wings after i.t. injection were also significantly lower compared with WT ZIKV (Fig. 2h). Furthermore, only 10% of saliva samples collected from xrRNA1′ mutant-infected mosquitoes 14 dpi were ZIKV-positive, whereas 35% of saliva samples were positive for the WT virus-infected group (Fig. 2g). Notably, no virus was detected in any saliva samples collected from mosquitoes infected with the xrRNA2′ mutant (Fig. 2g, h).

Collectively the results of infection and transmission analyses in mosquitoes inoculated via blood feeding or by i.t. injection indicate that deficiency in sfRNAs results in reduced ability of the virus to establish productive infection and to be transmitted.

**sfRNA-deficient ZIKV is capable of infecting salivary glands.** A previous study on WNV transmission suggested that sfRNA is required for penetration of the salivary gland barrier and infection of the salivary glands[17], which was based on reduced viral titres in saliva for an sfRNA1-deficient mutant. To determine if substantial reduction in sfRNA-deficient ZIKV mutants in saliva is due to the impaired ability of the viruses to infect salivary glands, histological sections were prepared from mosquitoes inoculated via i.t. injection and subjected to immunofluorescent detection of ZIKV NS1 protein (Fig. 3a–c). The results demonstrated that both sfRNA-deficient mutants had similar patterns of virus infection to the WT virus with strong staining of the viral protein in the head and thorax region (Fig. 3a, c), including the salivary glands (Fig. 3a). Intense staining of salivary glands was apparent even for the xrRNA2′ virus despite the infectious virus not being detected in any saliva samples for this mutant (Fig. 2g, h). Therefore, the absence of the virus in the saliva of xrRNA2′-infected mosquitoes

is not due to the inability of virus to overcome salivary gland infection barrier and infect salivary glands. A plausible hypothesis would be that the lack of virus in saliva of these mosquitoes is due to decreased secretion of the virus into the salivary gland ducts.

**Production of sfRNA prevents upregulation of *Caspase*-7 gene.** To identify host molecular processes targeted by ZIKV sfRNA, we assessed how production of sfRNA affects expression of *Ae. aegypti* genes by performing transcriptome-wide gene expression profiling of mosquitoes infected with WT ZIKV and the xrRNA2′ mutant. The xrRNA2′ mutant was selected due to its higher attenuation in vivo (Fig. 2g, h). The experiment used mosquitoes infected via i.t injection as it resulted in a 100% infection rate (Fig. 2g) and consistent viral titres were achieved within the experimental groups (Fig. 2h). In addition, qRT-PCR analysis for viral RNA also demonstrated similar viral RNA levels for WT and xrRNA2′ infections (Fig. 2f). RNA-Seq analysis performed at 10 dpe demonstrated that infection with WT ZIKV led to significant upregulation of 503 genes and downregulation of 107 genes, while infection with xrRNA2′ mutant significantly increased expression of 426 and decreased expression of 32 genes (Fig. 4a, b, Supplementary Fig. 4, Supplementary Data 1). Consistent with previous observations[24], WT ZIKV induced expression of immune effectors, including antimicrobial peptides (holotricin, gambicin and dipterin), phenol oxidases, clip domain serine proteases and leucine-rich immune proteins, while inhibiting the expression of defensins (Supplementary Data 1, Supplementary Figs. 4 and 5). xrRNA2′ mutant ZIKV had a similar effect on expression of these genes to WT virus except *holotricin*-3, which had a lower level of expression than for WT-infected mosquitoes, and *gambicin*, which was induced more strongly, albeit not significantly upon infection with the mutant virus (Supplementary Data 1, Supplementary Figs. 4, 5). Notably, infection with either virus caused only minor changes to the expression of genes involved in antiviral signalling pathways including Toll (Supplementary Fig. 5A, Supplementary Data 1).

Statistical comparisons of differentially expressed genes (DEGs) between mosquitoes infected with WT and xrRNA2′ ZIKV demonstrated that 50 genes exhibited a significantly different responses between WT and mutant virus (Fig. 4c, Supplementary Data 1). Notably, the gene encoding for the pro-apoptotic enzyme, caspase-7, was induced only by infection with the xrRNA2′ mutant and not by infection with the WT virus (Fig. 4c). This was further confirmed by qRT-PCR of RNA from individual mosquitoes infected within an independent inoculation experiment (Fig. 4d). qRT-PCR also demonstrated that infection with the xrRNA1′ mutant induced *caspase-7* expression, although the effect was less profound compared with the infection

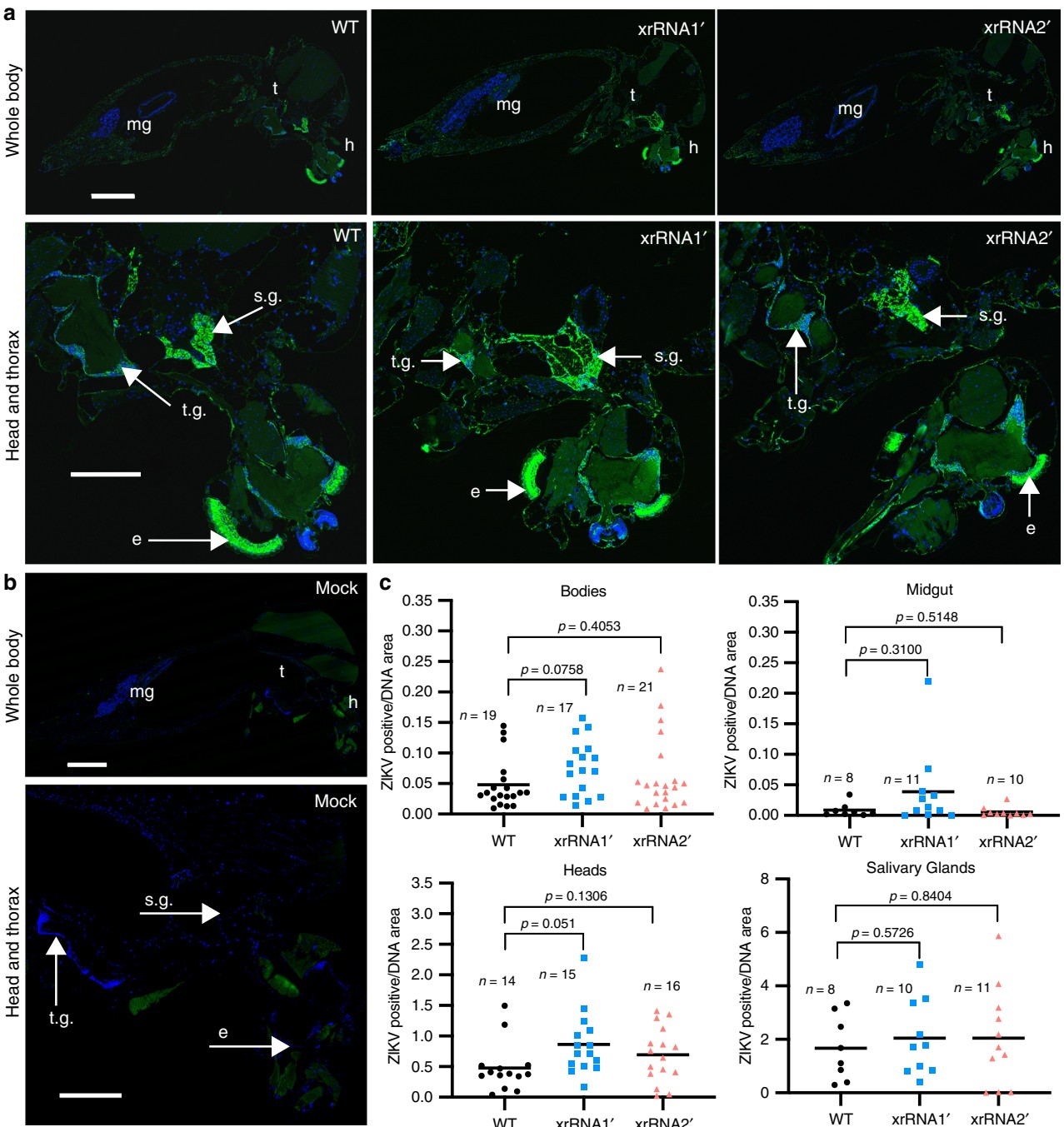

**Fig. 3 sfRNA-deficient ZIKVs infect mosquito salivary glands. a** Immunofluorescent detection of ZIKV NS1 (green) in histological sections of mosquitoes inoculated via i.t. injection and collected at 10 dpi. Bottom panels show magnified images of the major sites of viral replication in the head and thorax. Blue pseudo colour indicates for DAPI-stained nuclear DNA; mg, midgut; t, thorax; h, head; t.g., thoracic ganglia; s.g., salivary glands; e, eye. **b** Immunofluorescent staining of Mock-infected mosquitoes. **c** Quantification of ZIKV-infected areas in organs and tissues of mosquitoes. Statistical analysis was performed using Mann–Whitney U-test (independent comparisons, P-values are two-sided). Sample sizes, values for individual mosquitoes and the median (horizontal line) values for each group are shown. Images in (**a**) and (**b**) are representative microphotographs of 17–21 independent mosquitoes that showed similar results. Scale bars in (**a**) and (**b**) are 500 μm (whole body) and 250 μm (head and thorax), applicable to all images in corresponding sets.

with the xrRNA2′ mutant (Fig. 4d). Expression of other caspases was not affected by infection by either virus (Supplementary Data 1, Supplementary Fig. 5d). We also did not observe a differential response to infection in expression of genes involved in the RNAi pathway (Supplementary Fig. 5c). Gene ontology (GO) enrichment analysis of DEGs demonstrated that mosquitoes infected with WT virus had higher expression of genes involved in regulation of autophagy and genes related to

mitochondrial ATP synthesis (Fig. 4e). STRING analysis of these DEGs revealed that they encode for interacting components of the multi-protein complex of the mitochondrial respiratory chain, thus further indicating the biological relevance of the corresponding GO group (Fig. 4f).

Collectively, the results of gene expression analysis demonstrate that sfRNA does not affect the expression of genes involved in the IMD/Toll/Jak-STAT signalling but alters expression of

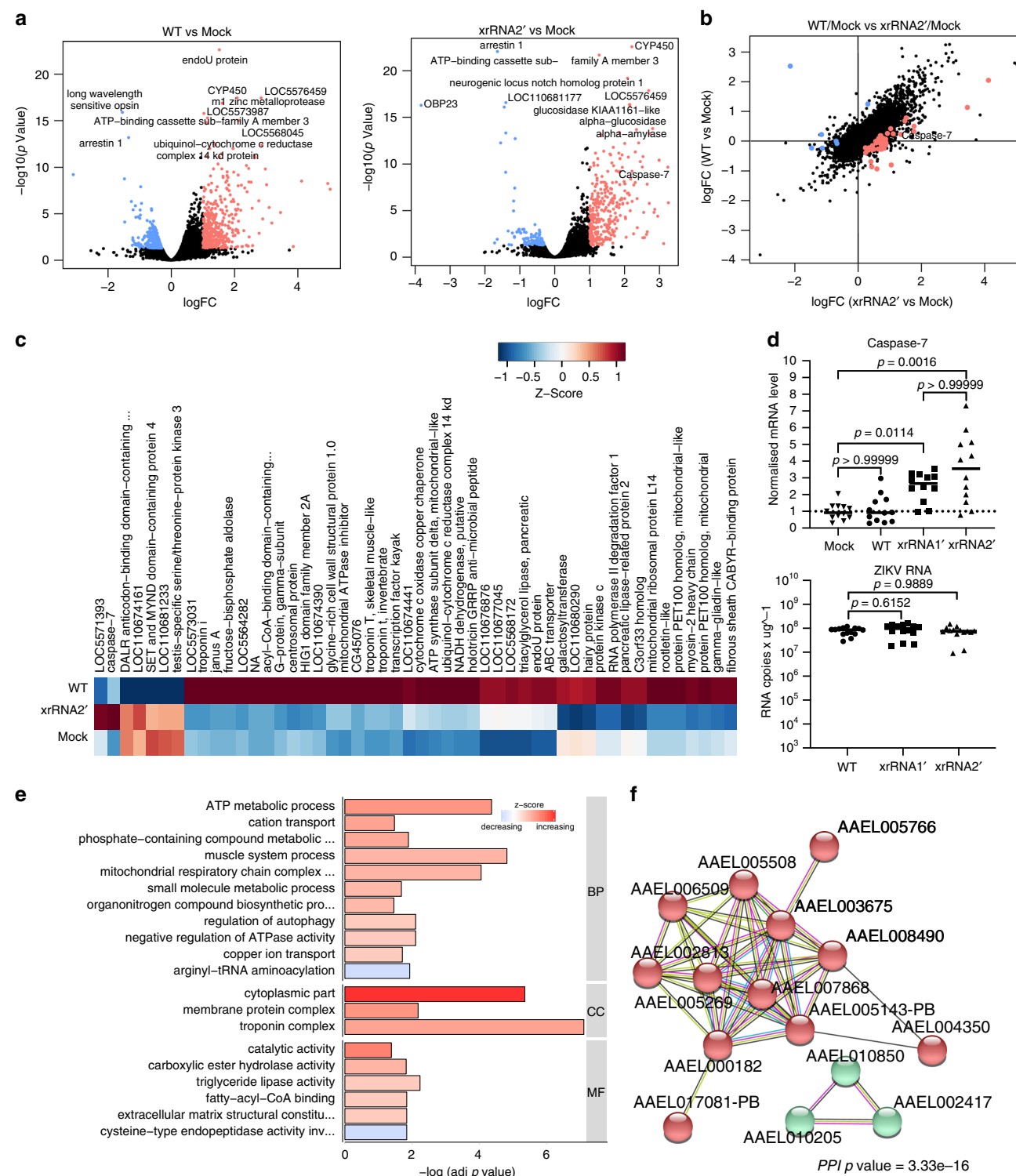

genes that directly (*Caspase*-7, autophagy-related genes) or indirectly (respiratory chain genes) control cell death. Therefore, we hypothesised that production of sfRNA may modulate apoptosis.

**ZIKV sfRNA inhibits apoptosis in infected mosquito tissues.** To determine if production of sfRNA affects apoptosis, histological sections of i.t. injected mosquitoes infected with WT, xrRNA1′ and xrRNA2′ viruses were subjected to TUNEL staining for apoptotic nuclei. The TUNEL assay showed that infection

with mutant viruses induced more apoptosis in infected mosquito tissues than WT virus infection (Fig. 5a). Quantification of apoptosis by calculating the ratios of TUNEL positive areas to DAPI-stained areas revealed that mosquitoes infected with mutant viruses had higher levels of apoptosis than WT virus in their bodies (Fig. 5a, b). The mutant viruses also induced more apoptosis in the salivary glands (Fig. 5b). In order to further analyse apoptosis levels in the tissues of infected mosquitoes, we performed statistical comparison of TUNEL staining between the groups of mosquitoes infected with WT ZIKV and sfRNA-deficient mutants. In this comparison, the production of sfRNA is

**Fig. 4 Production of sfRNA alters gene expression in ZIKV-infected mosquitoes.** Total RNA was isolated at 10 dpi after i.t. injection at the same dose of virus as in Fig. 2. RNA was extracted from 3 pools with 10 mosquitoes in each and used for RNA-Seq. Sequencing reads were mapped to the reference genome of *Ae. aegypti* and counted. Resulting count matrix was used for statistical testing for the differentially expressed genes (DEG). **a** Significantly up-regulated (red) and downregulated (blue) mosquito genes upon infection with WT ZIKV and xrRNA2′ mutant virus compared with mock. **b**, **c** Genes exhibiting different response to infection with WT virus and xrRNA2′ mutant shown on a scatter plot of expression changes (**b**) and heat map with indication of gene names (**c**). **d** Expression of *caspase-7* in individual infected mosquitoes at 10 days after i.t. injection as measured by qRT-PCR. Expression values were determined using the $\Delta\Delta C_T$ method with normalisation to mRNA levels of the *PRL11* housekeeping gene. The bottom panel indicates the levels of viral genomic RNA in respective mosquitoes as determined by qRT-PCR with standard curve extrapolation. Statistical analyses were performed using two-sided Kruskal–Wallis test with Dunn's correction for multiple comparisons. Graphs show individual and median (horizontal line) values. **e** Enriched GO categories associated with DEGs shown in (**b**). BP-biological processes, CC-cellular components, MF-molecular functions. **f** Interactions between proteins encoded by genes shown in (**e**) as identified by STRING analysis. Red nodes are the components of mitochondrial respiratory chain, green nodes are skeletal muscle proteins. DEGs in (**a**–**c**) were considered significant if FDR-corrected *P*-values were smaller than 0.05.

the only factor that differs between the groups, whereas other factors that may induce apoptosis must act in the same way. Given the presence of individual extreme values in some groups that do not belong to the observed distributions, we used the ROUT method[25] to identify outliers prior to the analysis and excluded them from the statistical test. The statistically significant difference in the induction of apoptosis by mutant and WT viruses was observed in bodies and salivary glands (Fig. 5c). The difference between the mutants, however, was not statistically significant (Supplementary Fig. 6). Significant difference in TUNEL staining between sfRNA-deficient mutants and WT ZIKV was also revealed for the thoracic ganglia (Fig. 5c). However, we did not observe substantial infection in this tissue after exposure to an infectious blood meal (Supplementary Fig. 3B) and no evidence was found in the literature for its role in flavivirus transmission. Therefore, we believe that apoptosis in thoracic ganglia (also evident in mock-infected mosquitoes, Fig. 3 and Supplementary Fig. 6) is unlikely to influence ZIKV transmission or reflect the biological functions of sfRNA.

The results of immunohistological analysis demonstrate that mosquito tissues infected with sfRNA-deficient ZIKV mutants were more prone to virus-induced apoptosis, thus supporting our hypothesis that ZIKV sfRNA has an anti-apoptotic activity in mosquitoes.

**Caspase inhibitor counteracts sfRNA deficiency in vivo.** To test if induction of apoptosis associated with sfRNA deficiency is the determining factor for the attenuated phenotypes of xrRNA1′ and xrRNA2′ mutants, the effect of pan-caspase inhibitor Z-VAD-FMK on in vivo infection and transmission was assessed. Mosquitoes were co-injected i.t. with each virus and either Z-VAD-FMK or vehicle control (NC) (Supplementary Fig. 2e) and after 14 days of incubation viral titres were determined in their bodies and saliva.

Consistent with our previous observations (Fig. 2g, h), infectious virus was present in the bodies of all mosquitoes from the vehicle control groups with the mean titres of both sfRNA-deficient mutants being significantly lower compared with WT virus (Fig. 6a). Significantly less of the xrRNA1′ and xrRNA2′ mutant viruses compared with the WT virus were also present in the saliva of mosquitoes co-injected with the vehicle control (Fig. 6b). Inhibition of caspases recovered replication of both sfRNA-deficient mutants in the bodies, with median viral titres in Z-VAD-FMK-treated mosquitoes infected with mutants being comparable to the titres of WT virus (Fig. 6a). The most dramatic improvement was observed in xrRNA2′ mutant virus-infected and Z-VAD-FMK-treated mosquitoes in which median virus titres increased by 17-fold compared with untreated mosquitoes (Fig. 6a). Notably, inhibition of caspases also significantly increased transmission of sfRNA-deficient mutants, with both mutants being detected in the saliva of 55% of infected

mosquitoes (Fig. 6c). Again, the xrRNA2′ mutant showed the most dramatic improvement from the treatment with the inhibitor, with 55% of mosquitoes being virus-positive compared with only 5% of virus-positive (1 mosquito) in the vehicle control-treated group (Fig. 6c). Although treatment with caspase inhibitor increased replication and transmission of sfRNA-deficient viruses, it did not affect WT ZIKV (Fig. 6a–c). This indicates that the treatment specifically compensates for the loss of sfRNA function rather than making mosquitoes more susceptible to viral infection in general. Therefore, the results of caspase inhibition further confirm that in mosquitoes, sfRNA facilitates ZIKV infection and transmission by inhibiting apoptosis.

## Discussion

Herein, we investigated the role of sfRNAs in ZIKV transmission and provided a mechanistic explanation for its functional significance in virus replication in mosquitoes. First, we found that mutant virus deficient in production of both sfRNA species, sfRNA1 and sfRNA2, could not be recovered in mammalian cells. When we attempted to recover this mutant virus in mosquito cells, the mutation in xrRNA2 reverted back to the WT sequence, restoring sfRNA2 production. These findings indicate that sfRNAs are critical for productive infection of ZIKV in mammalian and mosquito hosts. Single mutants deficient in either sfRNA1 or sfRNA2, were successfully recovered, and replicated in mosquito cells at the levels comparable to WT virus. However, these mutants exhibited decreased infection, dissemination and transmission in mosquitoes, similar to what was previously reported for WNV[17]. This indicates sfRNAs are targeting host processes in vivo that are either non-functional or not critical for viral replication in cell lines.

Notably, xrRNA2′ virus exhibited a more attenuated phenotype in vivo than the sfRNA1-deficient mutant. When xrRNA2′ mutant virus was administered via infectious blood meals, the xrRNA2′ mutation was eliminated in all mosquitoes. In contrast, in mosquitoes inoculated via i.t. injection, which bypasses the midgut barrier, the xRNA2 mutation was retained. The xrRNA1 mutation, which allows production of only sfRNA2, was retained when the xrRNA1′ mutant virus was administered via either route. These results suggest that sfRNA2 may play a more important role than sfRNA1 in the establishment of infection in the midgut. We also noted that, although the elimination of the xrRNA2′ mutation recovered viral replication in mosquito bodies and allowed penetration of the midgut barrier, virus secretion into the saliva of these mosquitoes was still drastically reduced. This could be due to a possible delay in dissemination associated with the time needed to acquire the reverse mutation, pass the midgut replication bottleneck, and establish replication in salivary glands. We should also bear in mind that only approximately half of the viruses with a 'C' to 'G' mutation in xrRNA2 reverted back

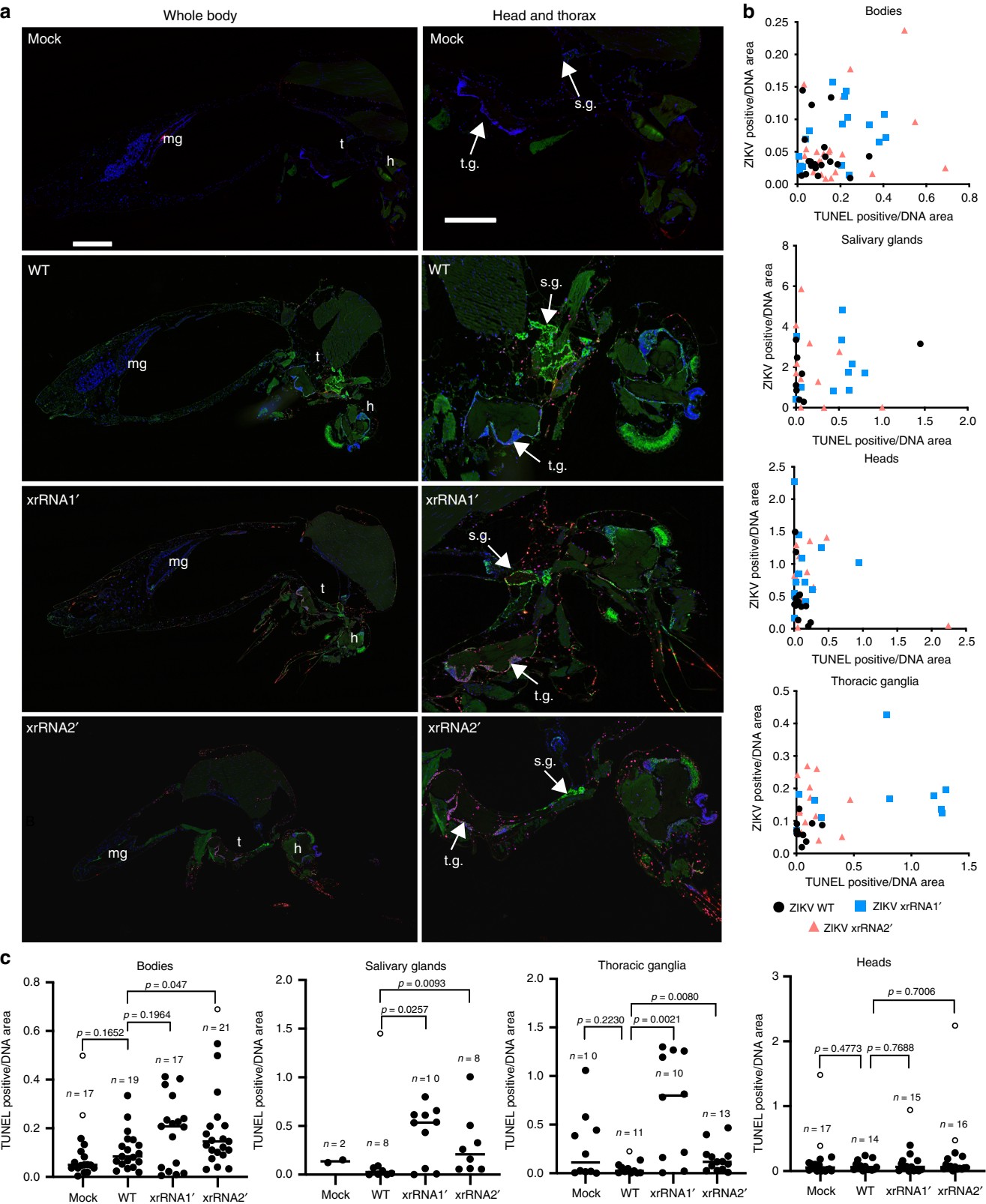

to the WT genotype (cytosine), while the other half contained adenine at this position. How the mutation into adenine and/or presence of mixed virus populations with cytidine and adenine may affect virus properties remains to be investigated.

A previous study with DENV demonstrated that shorter sfRNA isoforms (sfRNA2 and sfRNA3) were more important than sfRNA1 for virus adaptation to replication in mosquitoes[23]. Our results also point to the importance of sfRNA2 in flavivirus infections of mosquitoes as mutant ZIKV deficient in production of sfRNA2 appears to be more attenuated in vivo than the sfRNA1-deficient virus. Conversely, a mutation in xrRNA2 also reduced generation of sfRNA1 in infected mosquitoes, which

**Fig. 5 ZIKV sfRNA inhibits apoptosis in infected mosquito tissues. a** TUNEL staining of apoptotic nuclei (red) and ZIKV NS1 (green) in ZIKV-infected and mock-infected mosquitoes at 10 days after i.t. injection. Whole mosquito bodies and magnified organs with the major infection sites are shown on the separate panels. Blue pseudo colour shows DAPI-stained DNA; mg, midgut; h, head; t.g., thoracic ganglia; s.g., salivary glands. Images are representative microphotographs of 17–21 independent mosquitoes that showed similar results. Scale bars are 500 μm (whole body) and 250 μm (head and thorax), applicable to all images in corresponding sets. **b** Quantification of apoptotic and infected areas in organs of ZIKV-infected mosquitoes. All individual values are shown. **c** Statistical comparison of the apoptosis rate in the tissues of mosquitoes inoculated with WT and sfRNA-deficient ZIKV mutants shown in (**b**). Outliers (shown as open circles) were identified using ROUT method with $Q = 1\%$ (medium stringency) and cleaned data was analysed by two-sided Mann–Whitney U-test, each comparison was independent. Graphs show sample sizes ($n$), and individual and median values in each group.

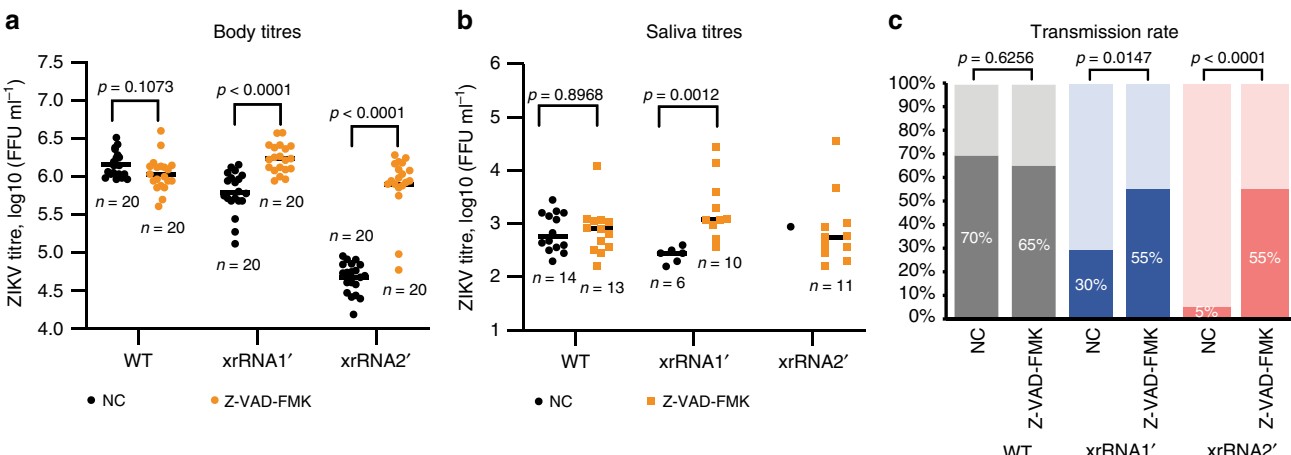

**Fig. 6 Effect of caspase inhibitor on in vivo replication and transmission of ZIKV.** Mosquitoes were i.t. injected with 200 nl of inoculums containing $10^4$ FFU/ml of each virus either with 625 μM of pan-caspase inhibitor Z-VAD-FMK or with vehicle control (NC). **a** ZIKV titres in mosquito bodies at 14 days after co-injection of viral inoculum with Z-VAD-FMK or vehicle control (NC). **b** Viral titres in saliva samples collected from caspase inhibitor-treated and untreated ZIKV-infected mosquitoes. **c** ZIKV transmission rate of WT and sf-RNA-deficient ZIKV by mosquitoes treated with Z-VAD-FMK or vehicle control (NC) determined as a percentage of ZIKV-positive saliva samples (darker shading on the graphs). All titres are determined by IPA on C6/36 cells. Statistical analysis is by independent Mann–Whitney U-tests (no multiple comparisons) (**a**, **b**) and chi-squared test (**c**). All P-values are two-sided, graphs in (**a**) and (**b**) show individual and median (horizontal lines) values, sample sizes ($n$) are indicated in the figure. NC, negative control (vehicle).

should be considered when assigning specific functions for sfRNA1 and sfRNA2 based on the properties of the xrRNA2′ mutant. Previous studies with DENV showed that virus adaptation to replication in mosquitoes involves accumulation of mutations in xrRNA2 that impair production of sfRNA1, suggesting that xrRNA2 is functionally coupled with xrRNA1[26,27]. Our results indicate that similar coupling may also take place in the ZIKV 3′UTR, however, the biochemical mechanism of how this coupling occurs remains to be established.

Combined results of blood feeding and i.t. injection experiments performed in our study demonstrated the requirement of ZIKV sfRNA for viral transmission. In order to systemically infect the mosquito vector and enable transmission, flaviviruses must overcome tissue infection barriers associated with the midgut and the salivary glands[28]. A previous study of a WNV deficient in sfRNA1 showed that this mutant had reduced viral secretion into saliva of blood fed mosquitoes[17]. Based on this observation and the reduced titres of sfRNA1-deficient WNV in saliva of infected mosquitoes, the conclusion was drawn that WNV sfRNA1 is required for penetration of the salivary gland barrier and establishing infection in this tissue[17]. While our manuscript was in revision, the same group also reported decreased infection rates and saliva titres for sfRNA1-deficient ZIKV, which led the authors to conclude that ZIKV sfRNA1 is required for viral dissemination into salivary glands[29]. Our study yielded similar results regarding the transmission capacity of the sfRNA1-deficient ZIKV mutant and showed that secretion into saliva of the sfRNA2-deficient mutant was even more impaired. However, our data employing immunohistochemical staining of infected

mosquitoes, which provides direct visualization of infected tissues, clearly show that sfRNA1 or sfRNA2-deficient mutants are capable of disseminating into, and establishing infection in the salivary glands (Fig. 3). Therefore, we conclude that ZIKV sfRNA is not required for salivary gland infection, but rather for the production and/or secretion of the infectious virus particles into saliva.

The function of sfRNA in mosquitoes was initially suggested to be executed through inhibition of the RNAi response[14,15]. However, this has been questioned recently[17] and currently there is no commonly accepted opinion on the function of sfRNA in mosquito infection. Our results, including (i) reversion of xrRNA2′ in xrRNA1′2′ double mutant observed in RNAi-incompetent C6/36 cells and (ii) demonstration of comparable levels of replication exhibited by ZIKV mutants with impaired production of sfRNA1 or sfRNA2 in either RNAi-competent or RNAi-deficient cells, indicates that the RNAi pathway is unlikely to be a target of sfRNA relevant to ZIKV infection. We therefore focussed on other immune processes in mosquitoes that may be affected by sfRNAs.

RNAi together with apoptosis determine innate intracellular immunity in mosquitoes, while secretion of antimicrobial peptides, melanisation and a complement-like pathway mediate a humoral immune response[30]. Upon arboviral infection, the humoral response is triggered by Toll, IMD and JAK-STAT signalling cascades[31]. Recently, sfRNA of DENV2 has been shown to inhibit expression of genes related to the Toll signalling pathway in the salivary glands of *Ae. aegypti*[16]. However, our gene expression profiling revealed that production of sfRNA by ZIKV

does not affect expression of these genes. Transcriptome profiling also demonstrated that production of sfRNA prevents upregulation of *caspase-7* and increases expression of genes involved in autophagy and the mitochondrial respiratory chain (Fig. 4). Caspase-7 is one of the effector caspases in *Aedes* mosquitoes responsible for cell death[32]. The mitochondrial respiratory chain also contributes to control of apoptosis as it affects generation of reactive oxygen species and cellular ATP levels[33–35]. Autophagy is believed to counteract apoptosis and promote survival of infected mosquito cells via recycling of damaged molecules and organelles[30,36].

Apoptosis was previously shown to inhibit replication, delay virus dissemination into salivary glands and reduce titres in saliva of *Aedes* mosquitoes infected with Sindbis virus[37]. In infected tissues of *Culex pipiens*, apoptosis was observed during WNV infection and shown to reduce infection rates and limit viral dissemination[38,39]. Furthermore, induction of apoptosis and increased expression of *caspase-7* were associated with resistance of *Ae. aegypti* to DENV infection[40]. Herein, we found that impaired production of sfRNA by ZIKV is associated with higher levels of apoptosis in infected mosquito tissues (Fig. 5), which was in line with the observed upregulation of *caspase-7* and decreased expression of respiratory chain components during infection with sfRNA2-deficient virus. Moreover, we found that replication and transmission of sfRNA-deficient ZIKV mutants was restored by introduction of a caspase inhibitor (Fig. 6). This demonstrates that the apoptosis pathway is a biologically relevant target of sfRNA and is responsible for viral attenuation associated with sfRNA deficiency. Considering these results and previously reported importance of apoptosis in the restriction of flavivirus replication in mosquitoes, the anti-apoptotic function of sfRNA identified here can explain why sfRNA production facilitates viral replication in mosquitoes.

In conclusion, we demonstrated the requirement of sfRNAs for viral replication and transmission by mosquitoes. We also discovered that production of sfRNAs alters expression of mosquito genes that control apoptosis, leading to inhibition of programmed cell death in ZIKV-infected tissues. Finally, we propose an anti-apoptotic activity of sfRNA as the mechanism defining the role of sfRNA in ZIKV-mosquito host interactions.

## Methods

**Cell culture**. Female African green monkey (*Cercopithecus aethiops*) kidney fibroblasts cells (Vero, ATCC – CCL-81; Vero 76, ATCC – CRL-1587), *Aedes aegypti* larvae cells Aag2 (ATCC – CCL-125) and *Aedes albopictus* larvae cells C6/36 (ATCC – CRL-1660) were obtained from the ATCC. *Aedes albopictus* larvae cells RML-12[41] were a generous gift from Prof. Robert Tesh (UTMB, USA). Vero cells were cultured in Dulbecco's modified Eagle's medium (DMEM) supplemented with 5% (v/v) foetal calf serum (FCS). C6/36 and RML-12 cell lines were cultured in Roswell Park Memorial Institute 1640 medium (RPMI 1640) supplemented with 10% (v/v) FCS, 10 mM HEPES pH 7.4. Aag-2 cells were cultured in 1:1 (v/v) mixture of Schneider's *Drosophila* medium and Mitsuhashi & Maramorosch medium (Sigma, USA) supplemented with 10% (v/v) FCS. All culture media were supplemented with 100 µg/ml streptomycin, 100 U/ml penicillin, and 2 mM L-glutamine. Vero cells were incubated at 37 °C with 5% $CO_2$. Insect cells (C6/36, Aag2 and RML-12) were cultured at 28 °C in sealed containers. All cell culture media and reagents were from Gibco, USA, unless specified.

**Virus and infection**. Zika virus strain MR766 was obtained from the Victorian Infectious Diseases Reference Laboratory. The virus was passaged once in C6/C6 cells and viral titres were determined by plaque-assay in Vero 76 cells. The viral genome was sequenced and matched GenBank MR766 reference sequence KU955594. All infections were performed at the indicated Multiplicity of Infection (MOI) by incubation of cells with 50 µl of inoculum per cm² of growth area for 1 h at 37 °C. Inoculated cell lines were then maintained in the growth medium containing a reduced amount of FBS (2%) to prevent overgrowth.

**Immuno-plaque assay (IPA)**. ZIKV titres were determined by immuno-plaque assay[42]. Tenfold serial dilutions of cell culture fluids or mosquito homogenates were prepared in DMEM media supplemented with 2% FBS and 25 µl of each

dilution were used to infect $2 \times 10^4$ Vero 76 cells grown in 96-well plates. After 2 h of incubation with the inoculum, 100 µl overlay media was added to the cells and incubated at 37 °C in 5% $CO_2$. The overlay media contained one part X M199 medium (containing 5% FCS, 100 µg/ml streptomycin, 100 U/ml penicillin, and 2.2 g/L NaHCO₃) and another part 2% carboxymethyl cellulose (Sigma-Aldrich, USA). At 3 days post infection, cells were fixed with 100 µl/well of 80% acetone for 20 min at −20 °C and washed with PBS, thoroughly dried, blocked for 30 min with 150 µl/well of Clear Milk blocking solution (Pierce, USA), and incubated with 50 µl/well of cross-reacting 4G2 mouse monoclonal antibody to flavivirus envelope (E) protein diluted in 1:100 for 1 h, followed by 1 h incubation with 50 µl/well of 1:800 dilution of goat anti-mouse IRDye 800CW secondary antibody (LI-COR, USA). All antibodies were diluted with Clear Milk blocking buffer (Pierce) and incubations were performed at 37 °C for 1 h. After each incubation with antibody, plates were washed five times with phosphate buffered saline containing 0.05% Tween 20 (PBST). Plates were then scanned using an Odyssey CLx Imaging System (LI-COR) (42 µm; medium; 3.0 mm). Virus replication foci were counted using the Image Studio Lite software (LI-COR, USA) and titres were determined based on dilution factors and expressed as focus forming units per mL (FFU ml⁻¹).

**RNA isolation**. Viral RNA was isolated from cell culture fluids using the NucleoSpin RNA Virus Kit (Macherey-Nagel, Germany). Total RNA from cells and mosquitoes was isolated using TRIreagent (Sigma, USA). Mosquitoes (10 per sample) were homogenised in 1 ml TRIreagent for 5 min at 30 Hz using a Tissue Lyser II (Qiagen, USA). All extraction procedures were conducted according to the manufacturer's instructions.

**RT-PCR**. RT-PCR for detection of viral RNA was performed using SuperScript III One-Step RT-PCR Kit with Platinum Taq (Invitrogen, USA) according to the manufacturers' recommendations. Twenty-five microlitre reaction mixtures containing 5 µl of viral RNA or 500 ng of total RNA and 5 pmol of each primer were incubated under the following thermal cycling conditions: 60 °C for 15 min, 94 °C for 2 min followed by 30 cycles of 94 °C for 15 s, 60 °C for 30 s, 68 °C for 1 min with final extension at 68 °C for 5 min. PCR products were analysed by electrophoresis in a 2% agarose gel.

**Circular polymerase extension reaction (CPER)**. For generation of DNA fragments that can be assembled into infectious DNA, viral RNA was isolated from culture fluids of Vero cells infected with Zika MR766 at 3 dpi and used as a template for first strand cDNA synthesis with SuperScript IV reverse transcriptase (Invitrogen, USA) according to the manufacturer's recommendations. Each RT reaction contained 11 µl of viral RNA, and 2 pmol of the reverse PCR primer for the corresponding fragment. RNA was denatured for 5 min at 95 °C, and annealed to the primers at 65 °C followed by cDNA synthesis at 55 °C for 1 h. DNA was then removed from RNA–DNA duplexes by incubation of RT mixture with 1 µl of *E. coli* RNase H (NEB, USA) for 20 min at 37 °C. RNase H-treated cDNA was used as a template for PCR with PrimeStar GXL Polymerase (Takara, Japan) and primers listed in Supplementary Table 1 according to the manufacturer's recommendations. The cycling conditions were 3 min at 98 °C; 40 cycles of 10 s at 98 °C, 15 s at 55 °C, 4 min at 68 °C; and a final extension for 5 min at 68 °C. PCR products were then separated in 1% agarose gel and DNA was extracted from the gel using Monarch DNA Gel Extraction Kit (NEB, USA).

PCR products that corresponded to the 3′-end of the viral genome (fragment 4) were cloned into the SmaI digestion site of the pUC19 vector. Primers for mutagenesis (Supplementary Table 1) were designed using NEBaseChanger online tool (NEB, USA [https://nebasechanger.neb.com]) and mutagenesis was performed using the Q5 Site-directed Mutagenesis Kit (NEB, USA) according to the manufacturer's instructions. The resulting plasmid was used as a template for PCR amplification of mutated fragments using the same conditions as for amplification of viral cDNA fragments. All enzymes and DH5a E. coli competent cells were from NEB, USA. Plasmid isolation was performed using the Wizard Plus SV DNA Miniprep System (Promega, USA).

Assembly of the infectious viral cDNA was conducted using the circular polymerase extension reaction (CPER) assembly[18,20]. PCR fragments 1–3 were mixed with the UTR-linker fragment (containing CMV promoter for mammalian cells or OpIE promoter for insect cells) and either WT ZIKV$_{MR766}$ fragment 4 or one of the three mutated fragments 4 (Supplementary Fig. 1A,B). CPER mixtures contained 0.1 pmol of each DNA fragment and 2 µl of PrimeStar GXL DNA polymerase (Takara, Japan) in a total reaction volume of 50 µl. Buffer, MgCl₂ and dNTP concentrations were as recommended by the manufacturer. The cycling conditions were 2 min at 98 °C; 20 cycles of 10 s at 98 °C, 15 s at 55 °C, 12 min at 68 °C; and a final extension for 12 min at 68 °C.

CPER products (50 µl) were transfected directly into Vero (under CMV promoter) or C6/36 (under OpIE promoter) cells using Lipofectamine LTX Plus transfection reagent (Life Technologies, Inc.). Briefly, each CPER product was mixed with 3 µl of PLUS reagent and 100 µl of Opti-MEM, followed by addition of 12.5 µl of Lipofectamine LTX dissolved in 150 µl of Opti-MEM. DNA-lipid complexes were incubated for 5 min at room temperature before being added onto the cells grown to 80% confluence in the wells of 6-well plates. At 24 h after transfection, cell culture medium was replaced with fresh medium containing

2% FBS. At 5, 7, and 10 days post-transfection (dpt), cell culture supernatant containing passage (P0) viruses were harvested and virus titres were determined by IPA. Viable P0 viruses were then used to infect C6/36 cells at an MOI of 0.1. At 5 and 7 days post-infection (dpi) culture fluids were collected from infected cells and titres of P1 virus were determined.

**DNA sequencing**. All PCR products, plasmids and viruses generated in the study were analysed by Sanger sequencing. Sequencing was performed by Australian Genomics Research Facility (AGRF) located in Brisbane, QLD, Australia.

**Northern blotting**. Detection of sfRNA was performed by Northern blotting[43,44]. Total RNA (10–20 µg) was mixed with equal volume of Loading Buffer II (Ambion, USA) and denatured by heating at 85 °C for 5 min followed by incubation on ice for 2 min. Samples of denatured RNA were subjected to electrophoresis in 6% polyacrylamide TBE-Urea gels (Invitrogen, USA). Electrophoresis was performed for 90 min in 1x Tris-Borate-EDTA buffer pH 8.0 (TBE). Gels were stained with ethidium bromide to visualise rRNA (bottom panels in Figs. 1c and 2c) and documented on GelLogic 212PRO imager (Carestream, Canada). RNA was then electroblotted onto Amersham Hybond-N$^+$ nylon membrane (GE Healthcare, USA) for 90 min at 35 V in 0.5x TBE using the TransBlot Mini transfer module (Bio-Rad, USA) and UV-crosslinking at 1200 kDj/cm$^2$. Membranes were then pre-hybridised at 40 °C in ExpressHyb Hybridization Solution (Clontech, USA) for 1 h. The probes were prepared by end labelling 10 pmol of DNA oligonucleotide complementary to the sfRNA (Supplementary Table 1) with [γ-$^{32}$P]-ATP (Perkin-Elmer, USA) using T4 polynucleotide kinase (NEB, USA) and purified from unincorporated nucleotides by gel filtration on Illustra MicroSpin G-25 Columns (GE Healthcare, USA). Hybridisation was then performed overnight at 40 °C in ExpressHyb Hybridisation Solution (Clontech, USA). After hybridisation, membranes were rinsed, washed 4 × 15 min with Northern Wash Buffer (1% sodium dodecyl sulphate [SDS], 1% saline-sodium citrate [SSC]) at 40 °C and exposed to a phosphor screen (GE Healthcare, USA) overnight. Signal detection was then performed on Typhoon FLA 7000 Imager (GE Healthcare, USA).

**Virus growth kinetics**. The growth kinetics of ZIKV MR766 wild-type and mutant viruses were assessed in C6/36, RML-12 and Aag2 cells. Cells were seeded at $2 × 10^6$ cells per well in separate 6-well plates. Cells were then inoculated with wild-type or mutated ZIKV MR766 at an MOI of 0.1 by incubating for 1 h with 200 µl of virus inoculum. Incubations were performed at 28 °C, then inoculum was removed, cells were washed three times with PBS and overlayed with 2 ml of their relevant culture medium supplemented with 2% FBS. At time point zero, 100 µl of media was immediately harvested from the wells, and infected cells were then incubated for 7 days at 28 °C. Culture fluid samples (100 µl) were then harvested at 3, 5 and 7 days post infection and subjected to IPA to determine the virus titres, from which growth curves were plotted.

**Rearing, inoculation and analysis of mosquitoes**. *Aedes aegypti* mosquitoes were obtained from a colony housed at Public Health Virology, Forensic and Scientific Services (FSS), Brisbane, Australia. The colony was established from eggs collected from Innisfail, Australia in April 2017 and the F$_{3-5}$ generations were used for the experiments. All mosquito exposures were conducted in Biological Safety Level (BSL) 3 insectaries at FSS or QIMR Berghofer Medical Research Institute.

For blood feeding experiments, *Aedes aegypti* mosquitoes were hatched and reared in plastic trays (48 × 40 × 7 cm) containing 3 L of rainwater at a density of 300 larvae per tray. Larvae were fed ground TetraMin Tropical Fish Food flakes (Tetra, Melle, Germany) ad libitum. Pupae were transferred to a dish of water inside BugDorm cages (30 × 30 × 30 cm) for adult emergence. Adult mosquitoes were provided with cotton wool soaked with 10% sugar solution which was withdrawn 48 h prior to blood feeding. Females were aspirated from the cages and placed into 750 ml containers with gauze lids at a density of ≈100 per container.

Three- to five-day-old mosquitoes were allowed to feed on blood meals containing WT and mutant ZIKVs, or a non-infectious blood meal. Each ZIKV virus was mixed with defibrinated sheep blood (Serum Australis, NSW, Australia) and offered to the mosquitoes via an artificial membrane feeding apparatus and the artificial membrane. Mosquitoes were allowed to feed for a period of 1 h. Pre-and post-feeding inoculum samples were collected and frozen at −80 °C. Directly after feeding, the mosquitoes were anaesthetized using CO$_2$ and immobilized on wet ice. Engorged mosquitoes were transferred to gauze-covered containers and maintained in an environmental chamber (Hipoint Co., Kaohsiung, Taiwan) at 28 °C, 75% relative humidity, 12:12 h light:dark cycling with 30 min dawn/dusk periods for 7 or 14 days.

For sample collection, mosquitoes were anaesthetised and placed on ice, legs and wings were removed and placed into a 1.5 ml microfuge tube. The mosquitoes were then carefully transferred to double-sided tape on a Perspex plate. Mosquito saliva was collected by placing a 200 µl pipette tip charged with 10 µl of saliva collection fluid (10% FBS/10% sugar solution) over the proboscis of each female. The females were provided 20 min to feed and salivate into the collection fluid. The fluid containing mosquito saliva was expelled into a 1.5 ml microfuge tube.

For inoculation via i.t. injection, viral inoculums were diluted in Opti-MEM medium (GIBCO, USA), supplemented with 3% foetal bovine serum (FBS; In Vitro Technologies, Australian origin), antibiotics and antimycotics (GIBCO, USA) to produce comparable titres for inoculation. The control group (Mock) was injected with virus-free medium. In the experiment with caspase inhibitor, the virus-containing inoculums were pre-mixed with 1/32 Vol of 20 mM solution in 100% DMSO to a final concentration of 625 µM and injected intrathoracically. The control group (vehicle control) was injected with inoculum contacting 625 µM DMSO in OPTI-MEM containing FBS (3%), antibiotics and antimycotics. The vehicle control group was injected with the medium containing 1/32 Vol of DMSO.

Four- to five-day-old female mosquitoes were anesthetised with CO$_2$ then, placed on a refrigerated table and injected intrathoracically with 200 nL of wild-type ZIKV Uganda MR766, xrRNA1′ or xrRNA2′ mutant viruses, respectively, at $5 × 10^4$ FFU/ml. Pre- and post-inoculation samples were diluted 1:10 in GM and stored at −80 °C. Inoculated mosquitoes were housed in 750 mL gauze-covered polypropylene containers and incubated at 28 °C, high relative humidity and 12:12 h light:dark within an environmental cabinet. Mosquitoes were provided with unrestricted access to 15% honey water. After 10 or 14 days of incubation bodies, legs/wings and saliva were collected separately to assess infection, dissemination and transmission for each mosquito, respectively. Bodies and legs/wings were separately placed into 2 mL U-bottom tubes containing 1 mL of OptiMEM medium + 3% FBS and a 5 mm stainless steel bead (Qiagen, USA). Saliva was collected into a capillary tube containing ~20 µl of GM supplemented with 20% FBS. Saliva expectorates were expelled into 600 µl of OptiMEM medium + 3% FBS, and along with the bodies and legs/wings were stored at −80 °C. Samples containing bodies and legs/wings were homogenised in a Tissue Lyser II (Qiagen, USA), and briefly centrifuged to remove chitinous debris. All samples, including saliva, were then filtered through a 0.2 µm syringe filter (Pall Corporation, Ann Arbor, MI). Virus titres were then determined in filtered homogenates using immunofluorescent plaque assay.

**Next-generation sequencing**. RNA samples from ten pooled *Aedes aegypti* mosquitoes infected with WT or xrRNA2′ mutant Zika MR766 or mock-infected were subjected to ribosomal RNA depletion using Ribo-Zero Gold rRNA Removal Kit (Illumina, USA) followed by library preparation using TruSeq 2 Library Preparation Kit (Illumine, USA). Three biological replicates (pools of mosquitoes) were used in the experiment. Libraries were sequenced on an Illumina HiSeq4000 instrument generating paired-end 75 bp reads. Image analysis was performed in real time by the HiSeq Control Software (HCS) vHD 3.4.0.38 and real-time analysis (RTA) v2.7.7, running on the instrument computer. RTA performs real-time base calling on the HiSeq instrument computer. The Illumina bcl2fastq 2.20.0.422 pipeline was then used to generate the sequence data. Library preparation, sequencing, and data acquisition were performed by the University of Queensland Genomics Facility and Australian Genomics Research Facility (AGRF). Quality control of raw sequencing data was performed using FastQC software v.0.72. Data was then trimmed to remove PCR primers, adapters and short reads using TRIMMOMATIC v.0.36.4 with the following settings: ILLUMINACLIP: TruSeq3-PE:2:30:10 LEADING:32 TRAILING:32 SLIDINGWINDOW:4:20 MINLEN:25 and subjected to another quality analysis with FastQC. Trimmed reads were mapped to the *Ae. aegypti* genome v.5.1[45] using HISAT2 v.2.1.0 allowing two mismatches. Feature count was performed using HTSEQ v.0.9.1 with counting mode set to "Union", strand to "Reverse", feature type was "exone" and ID attribute was GeneBank ID. Genome FASTA and GFF3 files were obtained from NCBI Gene Bank. RNA-Seq data generated in this study is available in Gene Expression Omnibus database with accession number GSE131827.

For viral amplicon sequencing 3′UTR fragments were amplified using ZIKV 3′UTR-Seq primers (Supplementary Table 1) and SuperScript III One-Step RT-PCR Kit with Platinum Taq (Invitrogen, USA) according to the manufacturer instructions. The cycling conditions were 15 min at 60 °C, 2 min at 95 °C; 30 cycles of 15 s at 95 °C, 30 s at 55 °C, 30 s at 68 °C; and a final extension for 5 min at 68 °C. Amplicons were purified with Monarch PCR & DNA Clean-up Kit (NEB, USA) and barcoded libraries were prepared using Ligation Sequencing Kit with PCR Barcoding Expansion 1-12 (Oxford Nanopore Technologies, UK). Sequencing was conducted on MinION sequencing device (Oxford Nanopore Technologies, UK) until the depth of 250k quality reads was reached. Base calling and demultiplexing was performed using Oxford Nanopore MinKNOW software (release 10/05/2018). Adapter sequences were then removed, and reads were quality-trimmed using Porechop v.0.2.3 (OmicX, France). Reads were mapped to the reference sequence of ZIKV$_{MR766}$ genome using Bowtie2 v.2.3.4.3, mapping was visualised and percentage of nucleotides in the positions of interest were determined using Integrative Genomics Viewer v.2.8.1 (Broad Institute, USA).

**Bioinformatic analysis of RNA-Seq data**. A table of raw read counts was prepared and annotated with gene names matching gene IDs using R-script. Annotations including gene names were obtained from VectorBase release VB-2018-10 using BioMart tool and GeneBank and combined. Differential gene expression analysis was performed using edgeR v.3.24.0. The normalisation method was TMM with Robust=TRUE, the likelihood ratio test (LRE) was applied to the contrasts WT-Mock (Supplementary Fig. 3A), xrRNA2′-Mock (Supplementary Fig. 3B) and (WT-Mock)-(xrRNA2′-Mock) (Fig. 3b, c). Genes were considered differentially

expressed if FDR-corrected $P$-values were <0.05. Gene expression data were plotted using ggplot2 v.3.1.0 and gplots v.3.0.1 with colour pallets generated by RColor-Brewer v.1.1-2.

Gene ontology analysis was performed using the R-package TopGO v.2.34.0 and GO annotations obtained from VectorBase. Fisher's exact test was applied. GO enrichment data was then combined with expression data, z-scores were calculated and results were plotted using the R package GOplot v.1.0.2[46]. Protein–protein interactions were identified using STRING v11.0 analysis. STRING analysis reconstructs the interconnection network taking a supplied list of genes to generate nodes and supplies edges that represent protein–protein associations from a database of published interactions. It also calculates a protein–protein interaction $P$-value (PPI $P$-value) based on how the number of interactions among the proteins in provided list compares to what would be expected for a random set of proteins of similar size, drawn from the genome. This enrichment indicates whether proteins are at least partially biologically connected as a group. STRING analyses were performed using the following settings: edge meaning vas "evidence", first shell was "<5", second shell was "none", active interactions sources were text mining, experiments, databases, co-expression, neighbourhood, gene fusion and co-occurrence.

**Quantitative RT-PCR (qRT-PCR)**. Total mosquito RNA (1 μg) was used to produce cDNA with qScript cDNA SuperMix (Quantabio, USA) according to manufacturer's instructions. cDNA was diluted 1:10 and 5 μl of cDNA solution were used as template for qRT-PCR using SYBR Green PCR Master Mix (Applied Biosystems, USA). PCR was performed in 20 μl of a reaction mix containing 10 pmol of forward and reverse PCR primers for each gene (Supplementary Table 1). Reactions were performed under the following cycling conditions: 95 °C for 5 min and 40 cycles of 95 °C for 5 s and 60 °C for 20 s, followed by melting-curve analysis using QuantStudio 6 Flex Real Time PCR Instrument (Applied Biosystems, USA). Gene expression levels were normalised to $RPL11$. Viral genomic RNA levels were determined using standard curve approach by comparing Ct values of the samples to Ct values observed in amplification of serial dilutions ($10^2–10^8$ copies/reaction) of a PCR-amplified and purified ZIKV genomic fragment. For each experiment, RNA from three biological replicates was used and PCR amplification of each cDNA sample was performed in triplicate. Negative controls were included for each set of primers.

**Immunofluorescent histological analysis and quantification**. Immunohistological analyses were conducted on 24 mosquitoes from each experimental and control group representing four independent infection batches. Samples of mosquitoes infected with ZIKV WT and mutant strains were dual stained by TUNEL Fluorescence in-situ hybridisation (FISH) against double stranded DNA breaks and immunofluorescence analysis (IFA) against ZIKV. This was achieved by applying the DeadEnd TUNEL Fluorometric System (Promega, USA) using a procedure modified to incorporate ZIKV IFA, as follows. Mosquitoes were thawed following storage at −80 °C, fixed in 4% paraformaldehyde/0.5% Triton X at 4 °C overnight and transferred to 70% ethanol. Mosquitoes were embedded in paraffin using standard histological procedures. Sections (3–4 μl) were cut and fixed to adhesive slides (Superfrost Plus, Menzel) and air-dried overnight at 37 °C. Samples were washed in 0.85% NaCl for 5 min at room temperature and Phosphate buffered saline (PBS) for 5 min at room temperature. The tissue sections were fixed in 4% methanol-free buffered Paraformaldehyde for 15 min at room temperature before washing samples well in reverse osmosis (RO) purified water at room temperature. One hundred and fifty microlitres of freshly prepared proteinase K (20 μg/ml in 0.05 M Tris-HCl, 0.01 M $CaCl_2$) for 10 min at room temperature. Slides were washed in two changes of RO water in a Coplin jar for 2 min each wash. The solution was removed before sections were incubated in 100 μl of Equilibration buffer for 10 min. The Equilibration buffer was removed. The slides were placed in a pre-heated humidified chamber placed in a 37 °C water bath. Sufficient volume of freshly prepared rTdT incubation buffer was added to cover the sections (≈50 μl) and a plastic coverslip was applied. The slides were incubated in the chamber for 60 min. The plastic coverslip was removed, and the reactions terminated by immersing slides in a Coplin jar containing 2× SSC for 15 min at room temperature. To remove unincorporated fluorescein-12-dUTP, samples were washed three times by immersing the slides in fresh PBS for 5 min at room temperature. This was repeated twice more for a total of three washes to remove unincorporated fluorescein-12-dUTP. The sections were incubated in Background Sniper solution (Biocare Medical, Walnut Creek, CA, USA) plus 1% BSA for 15 min to inhibit nonspecific antibody binding. Excess Background Sniper was removed and sections were incubated in 150 μl of primary antibody (undiluted 4G4 anti-Flavivirus NS1 monoclonal hybridoma supernatant[47]) overnight in a humidified chamber at room temperature. Sections were washed in three changes of phosphate buffered saline plus 0.05% Tween 20 (PBST). Sections were incubated in Alexafluor-555 conjugated donkey anti-mouse secondary antibody diluted 1:300 in PBST for 90 min at room temperature, after which they were washed three times in PBST. Sections were counterstained with DAPI for 5 min and washed three times in PBS before being mounted using Dako fluorescence mounting solution.

Slides were scanned using an Aperio ScanScope Fl slide scanner (Aperio Techologies, Vista, CA, USA) at a magnification of 20×. DAPI excitation was at 345 nm and emission collection at 455 nm with 0.1 s exposure. Fluorescein

excitation was at 495 nm and emission at 519 nm with a 0.32 s exposure. AlexaFluor® 555 excitation was at 555 nm and emission at 565 nm with a 0.32 s exposure. DeadEnd Tunel staining, ZIKV and DNA were false-coloured red, green and blue, respectively, in subsequent images. The staining density of Fluoroscein (DeadEnd TUNEL) and AlexaFluor® 555 (ZIKV) relative to DAPI (nuclear DNA) within specific mosquito tissues was determined[48]. Briefly, the tissue areas were delineated by circumscribing regions representing specific mosquito tissues (whole body, head, midgut, thoracic ganglia and salivary glands) using ImageScope Viewer software (Aperio). The area of positive pixels for each fluorophore (DAPI, Fluorescein and AlexaFluor® 555) within the regions was calculated using the Positive Pixel Area FL algorithm for each channel using the software eSlide Manager (Aperio) and ImageScope. For each tissue region, the areas of positive pixels for Fluorescein and AlexaFluor® 555 were then divided by the area of positive pixels for DAPI to give an approximate estimate of the proportion of cells positive for DeadEnd TUNEL and ZIKV, respectively. Each signal quantification involved correction for the background fluorescence. Immunohistological analyses were conducted on 24 mosquitoes from each experimental and control group representing four independent infection batches.

**Statistical analysis**. Statistical analyses were performed using GraphPad Prism Software v.7.0 or v.8.0, Microsoft Excel 360 and R v.3.5.0.

**Reporting summary**. Further information on research design is available in the Nature Research Reporting Summary linked to this article.

## Data availability
All data are available within the article, supplementary information or source data file. RNA-Seq data generated in this study are available in Gene Expression Omnibus database with accession number GSE131827. Raw data underlaying the results shown in Figs. 1b, c, 2a–h–h, 3c, 4d, 5b, c, 6a–c, supplementary Figs. 1B,D, 2A,B,E, 3C, 5D and 6A-D are provided in Source Data file.

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

## Acknowledgements

We acknowledge the excellent histology support of Clay Winterford (QIMR Berghofer) and thank Prof Scott Ritchie (JCU Cairns) for providing *Ae. aegypti* eggs from Innisfail, Australia. Work was funded by NHMRC grant APP1127916 to A.A.K. and Australian Infectious Diseases Research Centre's seed grant to R.H. and G.D. A.A.K. is the Senior Research Fellow with the NHMRC (APP1059794).

## Author contributions

A.S and A.A.K.: conceptualisation, experiment design, manuscript preparation; A.S., L.E.H., F.J.T., S.H.-M., M.F., A.A.A., N.Y.G.P., A.f.vhH., D.J.J.S. and X.Y.S.: experiments; A.S., L.E.H. and Y.X.S.: data analysis; A.S.: bioinformatics, R.A.H.: key reagents, A.A.K., R.A.H., G.J.D. and A.F.vdH.: project supervision.

## Competing interests

The authors declare no competing interests.
