## [Peer Review File · Nature Communications]

Reviewers' Comments:

Reviewer #1:

Remarks to the Author:

Flaviviruses produce a subgenomic RNA (sfRNA) due to Xrn-1 mediated degradation of viral genomic RNA and stalling on nuclease resistant structures in the 3'UTR (xrRNA structures). The function and mechanism of action of sfRNA has been subject to several studies, but remains incompletely understood, especially in mosquitoes. In this manuscript, Slonchak et al generate Zika mutants deficient in either the longer or shorter sfRNA species (sfRNA1 and 2) and analyze their replication and transmission potential in the vector mosquito *Aedes aegypti*.

The main findings are that i) Zika mutants defective in either sfRNA1 or 2 have replication defects in vivo, yielding lower infection rates in the saliva. No defect was observed in cells. ii) Mutants defective in both sfRNA1 and 2 are non-viable in both mammalian and insect cells, suggesting that it is crucial to retain at least one of the xrRNA structures. iii) The replication defect in vivo does not seem to be due to defects in RNAi suppression. iv) Increased expression of Caspase 7 is observed in infection with sfRNA2 mutant, but not wildtype virus (sfRNA1 mutant was not tested). In accordance, v) increased apoptosis is observed in mosquitoes infected with sfRNA defective viruses.

These results thus confirm the importance of sfRNA for mosquito infection and – by extension – vector competence (previously demonstrated for West Nile virus in *Culex* mosquitoes), propose a mechanism of action of sfRNA in vivo, suggest that apoptosis is important determinant for vector competence, and provide an explanation for the conservation of sfRNA in insect specific flaviviruses.

This study is well-designed and overall convincing. The mechanism by which sfRNA specifically affects caspase 7 expression remains to be established, but seems outside the scope of this manuscript.

Moreover, due to low vector competence, infection rates upon blood meals were low, and the authors thus inoculated mosquitoes by intra-thoracic injection. This is somewhat unfortunate, but a valid consideration that does not affect the importance of the results.

Main comments

- The observation that there is similar or even more ZIKA NS1 signal in different tissues in infections with the sfRNA mutants than in wildtype virus infections is unexpected given the lower viral loads and the reduced saliva infection rates. Also, I would not have expected prominent infection of the eye. A concern is that there is significant background/autofluorescence signal, which would not be unexpected for histological preparations. The authors should show a control staining of uninfected mosquitoes, and if present, control for background staining in their quantifications.
- Given that increased apoptosis is observed for both sfRNA1 and 2 mutants, it would be interesting to analyze Caspase-7 expression also for the sfRNA1 mutant in figure 3. Moreover, it would be of interest to analyze the expression of other caspases to confirm that the effect of sfRNA is specific to Caspase-7 (as suggested by RNAseq experiments).
- Figure 4 suggests that sfRNA inhibits apoptosis. However, the quantification shows TUNEL signal in the total tissue, not in the infected cells. Is it possible to quantify TUNEL in only NS1 positive cells? This would provide stronger support for a direct effect of sfRNA on apoptosis?
- The main conclusion that sfRNA suppresses apoptosis (e.g. in the title) is only supported by indirect evidence (caspase7 expression and apoptosis in infections with a sfRNA defective virus). To obtain more direct support for this conclusion, it is important to analyze whether sfRNA alone is sufficient to suppress apoptosis. I suggest an experiment in which Caspase expression and apoptosis is analyzed in cells transfected with sfRNA/an sfRNA encoding plasmid.

Minor comments

- The authors make a valid case to use injection to inoculate the mosquitoes. However, in this context, it does not seem appropriate to use the term dissemination (for infection of legs and wings); upon injection, the virus is directly introduced into the hemolymph, which will directly transport the virus to

all tissues. This is not similar to the process of dissemination from the midgut upon blood feeding. I suggest not using the term to avoid confusion.

- The stronger phenotype of the sfRNA2 mutant may be due to the higher expression of sfRNA2 over sfRNA1 (at least in vivo). It may be worth commenting on this.
- line 196: I am counting 50 genes in Figure 3C, not 45 as mentioned in the text. Please double-check the numbers/figure.
- Fig. 1C: It seems that the xrRNA1' mutant produces more sfRNA2 than wildtype virus, but this is difficult to assess when the amount of viral RNA is not known. Likewise, in Figure 2C, there seem to be much less sfRNA1 in the xrRNA2' mutant, which is likely due to the lower viral load. Please provide information about viral RNA levels in these samples, e.g. by qRT-PCR or by showing an uncropped gel image.
- Fig. 2A: please provide the statistics for the infection rates.
- At several instances, it was unclear to me what time point had been analyzed (e.g. Fig. 1C, RNA seq experiment of Figure 3). Please provide this information, where relevant.
- It would be useful to highlight Caspase 7 in the scatter plot of Figure 3B.
- Figure 4C, please also show the data for head and saliva, especially given the strong saliva phenotype in figure 2A.
- Figure S2. It is not clear to me what this figure is trying to show. Please explain.
- Figure S2B: typo 'befor' and 'titters'
- Figure S4: Why is Dicer-2 categorized under Jak-Stat? More generally, it may be of interest to show other RNAi genes as well.
- Line 413: I assume the authors are using a probe for sfRNA, not a miRNA.
- Supplemental tables: please define abbreviations. How are read counts reported?

Reviewer #2:

Remarks to the Author:

In the manuscript 'Zika virus noncoding RNA suppresses apoptosis and is required for virus transmission by mosquitoes' by Slonchak et al., the authors present their work on the role of ZIKV sfRNA1 and sfRNA2 during virus transmission by *Aedes aegypti* mosquitoes. The work is interesting, with novel aspects, and should be of interest to a large audience of virologists. However, I have a number of major concerns regarding the design of the study and interpretation of data.

1) I think the study would have been more informative if performed in mosquitoes that are more susceptible to oral bloodmeal infection. The I.T. injection bypasses the midgut and causes injury and may not be the best way to truly evaluate the role of sfRNA. Especially, since these mosquitoes are not as susceptible and may have other confounding factors that impact virus replication after injection. However, I understand that it may not be possible to avoid in the specific laboratory setting due to containment or import permit limitations. If possible, a simple comparative experiment using the three viruses and an infectious bloodmeal as additional data would be useful.

2) In addition, no titers were provided for bloodmeals and injections, please add these!

3) As described again in my line-by-line comments below, it appears that the readout for all infectious virus titrations is a focus forming assay on Vero cells. The author show early on that the double mutant is replication incompetent in Vero cells and the single mutants also appear attenuated (at least xrRNA2'). This would suggest that a titration on Vero cells is not necessarily going to represent what is actually happening in the mosquitoes. A focus forming assay on mosquito cells and RNA based titrations seem would be more appropriate. Especially since the authors later on mention that RNA levels in mosquito samples from wt and mutant viruses were comparable (in the RNAseq samples)

4) The conclusions drawn by the authors from their data do not always appear to be supported as outlined in my specific comments below.

5) The apoptosis results are a little bit confusing. It seems like the xrRNA2' mutant was generally the more attenuated mutant, but in Figure 4, the data on increased TUNEL staining in thoracic ganglia and salivary glands is much more convincing in the xrRNA1' mutant, which was only mildly attenuated in the mosquito. This may in fact suggest that apoptosis is not the mediator for the reduced transmission that was observed. Additional experiments would provide more concrete evidence for the hypothetical role of apoptosis in reducing ZIKV xrRNA mutant transmission (e.g. using an apoptosis inhibitor to try to overcome the block in ZIKV transmission). In addition, including xrRNA1' mutant samples in the transcriptomic analysis may help elucidate what is happening or if other pathways/mechanisms are involved.

Line-by-line comments:

Line 43: should be 'conductive to'

Line 61: Suggestion: '...and the less abundant shorter sfRNA-2...'

Line 67: The authors state: 'However, the exact role of sfRNA in arthropods is still unclear'. While the exact role is indeed unclear, the authors should highlight what is known and what potential roles for sfRNA have been proposed previously. For example, there are papers suggesting that Langkat virus and tick-borne encephalitis virus sfRNAs can inhibit RNA interference in ticks (Schnettler et al 2014), that dengue sfRNA may inhibit Toll signaling in salivary glands (Pompon et al. 2017), that WNV sfRNA1 is important for virus infection of the mosquito midgut and virus transmission (Goertz et al 2016) and that sfRNA has RNAi suppressor activity in mosquito cells (Schnettler et al. 2012; Moon et al. 2015). While some of these papers are cited later on, it would be useful to have a sentence or two here in the introduction providing some background on the various potential roles of sfRNAs in arthropods (or at least in mosquitoes). The introduction currently makes it sound as if this is the first study investigating this question.

English: please check throughout for article usage. The definite article (and sometimes indefinite article) is missing at times (e.g. line 72: '... ZIKV replication in the insect host...' or as mentioned above in line 62). In line 100 should be '... contained a reversion...'

Figure 1 legend: more info could be provided, e.g. source of structure (crystal structure? model? How was it modelled?), time post transfection in B, add at least 'by Northern blot' or similar for C, and if error bars represent just standard deviation in D. Just a bit more detail would be useful for the reader.

Lines 96-102: The authors here describe an experiment using C6/36 cells to grow an sfRNA1+2 deficient virus, yet the measure of infectious virus produced is an immune-assay on Vero cells. The authors previously show that Vero cells do not support replication of this virus. So even if the xrRNA1'2 ZIKV mutant was able to replicate in C6/36 cells and generated infectious virus it would not be infectious on Vero cells (just C6/36 cells). The fact that a reversion occurred here does also not prove that the xrRNA1'2 ZIKV mutant virus cannot replicate in C6/36 – in fact, some replication must have occurred for the mutation to arise, or alternatively the cDNA prep was not clean and contained a small amount of xrRNA1' with intact xrRNA2. I would suggest using C6/36 cells as the read-out for infectious virus, as well as measuring extracellular viral RNA over time, and providing information on how many times this experiment was conducted. Currently, the conclusion that 'Therefore, generation of at least one sfRNA isoform is required for ZIKV viability in mosquito cells even with dysfunctional RNAi.' Is not warranted.

Lines 114-121: this is repetitive and discussion. Also, the 'therefore' in line 114 is not a logical connection from the previous section where the generation two separate mutants for sfRNA 1 and 2 was already described. I would suggest to remove this section. In addition, my previous comments about replication of the double mutant in C6/36 cells negates a lot of the conclusions provided here without further evidence that the double mutant truly does not replicate in C6/36 cells.

Line 126: what were the titers used for bloodmeal and inoculation??

Figure 2: If the 'n=69' were moved into the middle of the graph, under the heading 'Blood feeding' it would be more obvious that it does not just refer to the wildtype bar, since it is currently situated right above that bar (same for the other panels). Presumably the 'n' was the same for each group (wt,

xrRNA1', xrRNA2'). In the legend for panel A, it says 'inoculated mosquitoes' – please change to 'mosquitoes exposed to ZIKV by bloodmeal with a titer of ...' or similar, for clarity. Please also provided i.t. inoculation titers for B. In the last sentence an extra space should be removed ('tis sues' should be 'tissues').

Line 131/132: This is a major drawback of this study. Why did the authors not try to obtain a more susceptible mosquito line? Are there specific reasons (containment, import etc?) why this wasn't possible?

Line 133ff: It should be noted somewhere in this section that sfRNA1 and sfRNA 3(?) were less abundant in the xrRNA2' mutant infected mosquitoes, which probably just correlates to lower viral RNA levels overall (since the blot is normalized to mosquito RNA not viral RNA). Clarifying this here would help the reader.

Lines 148-149: the authors talk of 'dissemination to the legs and wings'. This is not a useful measure or description for an i.t. inoculation which initiates a disseminated infection without crossing the midgut barrier. It is reasonable to keep the data in, if desired although it does not add much information), but the word dissemination in vector competence studies refers to dissemination throughout the body after crossing the midgut barrier and is not appropriate here. Please re-word.

Lines 150-158: please check/correct English mistakes in this section for clarity.

Line 173: the authors mention 'replication efficiency', however the viral titers from mosquito tissues were all obtained in Vero cells (presumably) where these mutants (especially xrRNA2', see Fig1B) are also attenuated. Other outputs such as viral RNA or a focus-forming assay using C6/36 cells could be useful to identify whether it is viral RNA replication, virus production or virus infectivity in Vero cells used to measure titers that's providing the (somewhat confusing) results. With the NS1 staining looking comparable, it is hard to imagine that viral RNA replication and infection of mosquito cells themselves are as significantly impacted as the titers insinuate.

Line 206: here the authors mention that RNA content was comparable between wt and xrRNA2' mutant viruses suggesting that virus spread through the mosquito was probably not impacted, but not detected in the earlier experiments due to the 'virus titer on Vero cells' readout.

Figure 4: The x-axes are all different in Figure 4B, making it very hard to compare data. In lines 225-228 the authors describe signs of apoptosis based on these data and I don't fully agree with the authors' conclusions (e.g. 'A limited number of mosquitoes had considerable signs of apoptosis in the head (Fig. 4B) with the only TUNEL-positive mosquitoes being those infected with mutant viruses' – there are wildtype data points that have some TUNEL staining, but the axis is very different to that for bodies, effectively reducing data resolution. Similarly, the authors state that 'Mutant viruses were also the only ones causing considerable apoptosis in salivary glands', when there is clearly at least one sample from wildtype virus that has significant TUNEL staining (more than any other mosquito). There are also xrRNA2' exposed mosquitoes without any virus in the salivary glands but significant TUNEL staining, which could indicate a confounding factor (or potentially TUNEL activation through extracellular factors). The data is in fact clearer for xrRNA1' mutants, which is counterintuitive considering these viruses are less attenuated compared to xrRNA2' mutants. It should also be noted that no uninfected control group was included. It would have been valuable to see TUNEL staining in uninfected mosquitoes for comparison.

Line 240: please don't use 'ZIKV life cycle' or mosquito stage in this context. It is a transmission cycle if you want to give it a name as such. The sentence should just be restructured to avoid these terms.

Line 264: The authors state: 'Thus, deficiency in sfRNAs appears to have a more profound effect on ZIKV infection than on WNV infection, although sfRNAs are clearly required for efficient replication and transmission in mosquitoes for both viruses'. I think it is worst repeating here why the authors think 'sfRNA appears to have a more profound effect on ZIKV...', i.e. because the authors used i.t. injection and still saw an effect. However, it is also hard to compare the results, since we don't know what would have happened after a successful infectious bloodmeal infection. Maybe the midgut aspect of the sfRNA is less pronounced in ZIKV, making the overall impact similar. I suggest elaborating a little bit here to be clear and so that only fair comparisons are made.

Methods:

Lines 311-313: please also provide the equivalent information/origin of RML-12 cells.

Line 326: the authors haven't mentioned 'vero76 cells' what are these? What does the 76 stand for? Please add them in the 'cell culture' section.

Line 436: Please add 'Innisfail, Australia,' for clarity. The choice of using a nearly refractory mosquito line for these experiments seems a bit odd, although I understand if it was a matter of mosquito strain availability.

Line 440: Please provide information on virus titers in the blood meal. This may provide insight into the low infection rates observed and is important for the reader. Please also provide information on the ratios of blood and virus stock, as well as how virus stock concentrations were normalized (e.g. using culture media??).

Line 513: I assume the authors mean '60°C for 20 s' not minutes here, please correct.

Reviewers' comments:

Reviewer #1 (Remarks to the Author):

This study is well-designed and overall convincing. The mechanism by which sfRNA specifically affects caspase 7 expression remains to be established, but seems outside the scope of this manuscript. Moreover, due to low vector competence, infection rates upon blood meals were low, and the authors thus inoculated mosquitoes by intra-thoracic injection. This is somewhat unfortunate, but a valid consideration that does not affect the importance of the results.

We are pleased to see that the reviewer found our study to be well designed and convincing. We have overcome the modest vector competence in blood meal infections by using higher viral doses. These new results support the conclusions from the i.t.-injection experiments and are presented in Figure 2 of the revised manuscript.

Our response to other comments of the reviewer are presented below:

Main comments

- The observation that there is similar or even more ZIKA NS1 signal in different tissues in infections with the sfRNA mutants than in wildtype virus infections is unexpected given the lower viral loads and the reduced saliva infection rates. Also, I would not have expected prominent infection of the eye. A concern is that there is significant background/autofluorescence signal, which would not be unexpected for histological preparations. The authors should show a control staining of uninfected mosquitoes, and if present, control for background staining in their quantifications.

In the revised manuscript we provide control staining of uninfected mosquitoes for experiments involving immunohistology (Fig 3B, 5A,C, S3A,B, revised manuscript). Although some background was evident on the sections of uninfected mosquitoes, the sites of infection were clearly distinguishable from autofluorescence as they have much higher brightness of staining. All image quantification analysis algorithms were confirmed to exclude background. We now specify this in the Methods section (lines 851-852 of the revised manuscript). We believe that similar levels of NS1 as well as similar levels of viral RNA despite the difference in the virus titres indicate that sfRNA production does not influence replication of viral RNA and translation of viral proteins, but rather affects virus secretion into saliva.

Infection in the eyes was previously reported for DENV (Platt K.B et al., 1997) and WNV (Girard Y.A. et al, 2004). However, as eye infection is unlikely to affect viral transmission, we did not investigate this further in this manuscript.

- Given that increased apoptosis is observed for both sfRNA1 and 2 mutants, it would be interesting to analyze Caspase-7 expression also for the sfRNA1 mutant in figure 3. Moreover, it would be of interest to analyze the expression of other caspases to confirm that the effect of sfRNA is specific to Caspase-7 (as suggested by RNAseq experiments).

We have now determined expression of Caspase-7 in mosquitoes infected with xrRNA1' mutant by qRT-PCR. We found that it was up-regulated comparing to WT-virus infection, although the effect was less profound than after infection with xrRNA2' mutant (Fig 4D in the revised manuscript). We have also determined expression of the caspase Dronc and found that it was not affected by infection with either virus (Fig S5D in the revised manuscript).

- Figure 4 suggests that sfRNA inhibits apoptosis. However, the quantification shows TUNEL signal in

the total tissue, not in the infected cells. Is it possible to quantify TUNEL in only NS1 positive cells? This would provide stronger support for a direct effect of sfRNA on apoptosis?

We agree with the reviewer that quantification of apoptosis in individual cells would be more informative, however, it is unfortunately not technically achievable as the available image quantification software cannot identify and isolate the boundaries of the individual cells.

- The main conclusion that sfRNA suppresses apoptosis (e.g. in the title) is only supported by indirect evidence (caspase7 expression and apoptosis in infections with a sfRNA defective virus). To obtain more direct support for this conclusion, it is important to analyze whether sfRNA alone is sufficient to suppress apoptosis. I suggest an experiment in which Caspase expression and apoptosis is analyzed in cells transfected with sfRNA/an sfRNA encoding plasmid.

We are not convinced that transfecting sfRNA alone into uninfected cells followed by induction of apoptosis by various treatments will provide more direct evidence of its role in inhibiting apoptosis. It should be noted that even those arboviruses that do induce apoptosis in mosquitoes *in vivo* don't do so in mosquito cell lines (Sim S., et al, 2014). This indicates that cultured mosquito cells (that originated from larvae and are immortalised) may be lacking the pro-apoptotic pathways relevant to arbovirus infection and therefore sfRNA function. Lack of the effect of individual sfRNA mutations on ZIKV replication that we observe in cells also suggests that this may be the case. Therefore, we have generated new data that supports our conclusion on antiapoptotic function of sfRNA during infection in mosquitoes which represents more physiologically relevant infection environment (see also response to reviewer 2). We have demonstrated that treatment of mosquitoes with caspase inhibitor Z-VMD-FAK recovers replication and transmission of sfRNA-deficient mutants, but doesn't affect replication and transmission of WT virus (Fig 6 of the revised manuscript). We believe this now provides direct support for our conclusion that inhibition of apoptosis is the mechanism of sfRNA action during infection.

Minor comments

- The authors make a valid case to use injection to inoculate the mosquitoes. However, in this context, it does not seem appropriate to use the term dissemination (for infection of legs and wings); upon injection, the virus is directly introduced into the hemolymph, which will directly transport the virus to all tissues. This is not similar to the process of dissemination from the midgut upon blood feeding. I suggest not using the term to avoid confusion.

We agree and have removed term "dissemination" from the description of the results obtained using i.t.-injections.

- The stronger phenotype of the sfRNA2 mutant may be due to the higher expression of sfRNA2 over sfRNA1 (at least *in vivo*). It may be worth commenting on this.

We have added discussion on this in lines 366-369 of the revised manuscript.

- line 196: I am counting 50 genes in Figure 3C, not 45 as mentioned in the text. Please double-check the numbers/figure.

We have corrected numbers in the text

- Fig. 1C: It seems that the xrRNA1' mutant produces more sfRNA2 than wildtype virus, but this is

difficult to assess when the amount of viral RNA is not known. Likewise, in Figure 2C, there seem to be much less sfRNA1 in the xrRNA2' mutant, which is likely due to the lower viral load. Please provide information about viral RNA levels in these samples, e.g. by qRT-PCR or by showing an uncropped gel image.

No RNA bands were cropped out the Northern blot image provided in this figure, however full-length viral genomic RNA is too large to enter the 6% polyacrylamide gel used in our experiments and thus cannot be detected by this method. Vero cells infected with all viruses had comparable levels of viral RNA because they had similar viral titres, as shown in Fig 1C (Fig 1B under the ribosomal RNA panel, revised manuscript). For the *in vivo* experiment (Fig 2E, revised manuscript) we have now quantified viral RNA using qRT-PCR as the reviewer requested (Fig 2F, revised manuscript). We found no difference in the levels of viral RNA between mosquitoes infected with the xrRNA2' mutant and other viruses. We describe these results in lines 189-193 of the revised manuscript.

- Fig. 2A: please provide the statistics for the infection rates.

In the revised manuscript we now provide statistical analyses for all infection, dissemination and transmission rates (Fig 2A,C,G, revised manuscript).

- At several instances, it was unclear to me what time point had been analyzed (e.g. Fig. 1C, RNA seq experiment of Figure 3). Please provide this information, where relevant.

This information was added to the legends of the respective figures.

- It would be useful to highlight Caspase 7 in the scatter plot of Figure 3B.

Caspase 7 is now highlighted (Fig 4A, revised manuscript).

- Figure 4C, please also show the data for head and saliva, especially given the strong saliva phenotype in figure 2A.

We have added these data to the manuscript (Fig. 5, revised manuscript)

- Figure S2. It is not clear to me what this figure is trying to show. Please explain.

This figure shows viral titres for each virus in the inoculums used for mosquito infection. It also contained a graphic outline of the experiment, which has been removed from the revised manuscript and replaced with deep sequencing data.

- Figure S2B: typo 'befor' and 'titters'

Corrected.

- Figure S4: Why is Dicer-2 categorized under Jak-Stat? More generally, it may be of interest to show other RNAi genes as well.

We showed Dcr2 in Jak-Stat category because Dicer-2 was shown to act as pattern recognition receptor for this pathway and trigger Jak-Stat response to WNV infection in *Culex* mosquitoes (Paradkar P.N. et al., 2014). In the revised manuscript we included a footnote into the legend of Fig

S5 (replacing Fig S4) with this explanation. We have also added a new panel for RNAi pathway genes.

- Line 413: I assume the authors are using a probe for sfRNA, not a miRNA.

Corrected.

- Supplemental tables: please define abbreviations. How are read counts reported?

This information has been added to the figure legends.

Reviewer #2 (Remarks to the Author):

The work is interesting, with novel aspects, and should be of interest to a large audience of virologists.

We thank reviewer for these positive comments.

However, I have a number of major concerns regarding the design of the study and interpretation of data.

1) I think the study would have been more informative if performed in mosquitoes that are more susceptible to oral bloodmeal infection. The I.T. injection bypasses the midgut and causes injury and may not be the best way to truly evaluate the role of sfRNA. Especially, since these mosquitoes are not as susceptible and may have other confounding factors that impact virus replication after injection. However, I understand that it may not be possible to avoid in the specific laboratory setting due to containment or import permit limitations. If possible, a simple comparative experiment using the three viruses and an infectious bloodmeal as additional data would be useful.

The strict biosecurity regulations in Australia make importation of foreign mosquito strains very difficult and therefore we had to work with a locally available mosquito strain. To overcome the low competence of the local strain, we exposed mosquitoes to the blood meal with a 100-fold higher viral dose than previously used. The experiment resulted in infected mosquitoes and yielded the same outcome as i.t.-injection (viral attenuation associated with sfRNA deficiency) and provided further support to our conclusions about the requirement of sfRNA for ZIKV transmission. These results are now presented in Fig 2A-D, Fig S2B, and Fig S3 of the revised manuscript.

2) In addition, no titers were provided for bloodmeals and injections, please add these!

The titres are now shown in Fig 6A and Fig S2A of the revised manuscript.

3) As described again in my line-by-line comments below, it appears that the readout for all infectious virus titrations is a focus forming assay on Vero cells. The author show early on that the double mutant is replication incompetent in Vero cells and the single mutants also appear attenuated (at least xrRNA2'). This would suggest that a titration on Vero cells is not necessarily going to represent what is actually happening in the mosquitoes. A focus forming assay on mosquito cells and RNA based titrations seem would be more appropriate. Especially since the authors later on mention that RNA levels in mosquito samples from wt and mutant viruses were comparable (in the RNAseq samples)

As we understand also from the comment to line 173, the reviewer may be confused by Fig. 1B of the original manuscript which shows less plaques in culture fluids of Vero cells transfected with infectious cDNA that encodes for xrRNA2' virus. This plate was actually a virus titration assay on

Vero cells to determine virus titres and also to illustrate that double mutant was not recovered. This was not meant to show attenuation of mutant viruses in Vero cells. To avoid the confusion this panel has been removed.

We also demonstrated that all viruses replicated to the similar titres in Vero cells upon infection at the same MOI (viral titres under Northern blot panels in Fig 1C of original manuscript and Fig 1B of revised manuscript). In the revised manuscript we also provide growth kinetics in Vero cells for all three viruses with by focus forming assay on C6/36 cells (Fig. S1D, revised manuscript), which further confirms lack of attenuation in Vero cells for either mutant. We respectfully disagree with the reviewer that RNA-based titration will be more appropriate than focus forming assay for infectious virus because this method also picks aberrant RNAs in defective infectious particles and RNA leakage from dying cells and generally considered to be less accurate.

Finally, when performing the experiment with injection of apoptosis inhibitor as suggested by the reviewer (below), we determined viral titres in mosquitoes by IPA (focus forming assay) on C6/36 cells. In the control groups for this experiment (injection with virus, but no inhibitor) we obtained similar results as we previously obtained with IPA on Vero cells. All these results demonstrate that the titres determined by IPA on Vero are biologically relevant.

4) The conclusions drawn by the authors from their data do not always appear to be supported as outlined in my specific comments below.

5) The apoptosis results are a little bit confusing. It seems like the xrRNA2' mutant was generally the more attenuated mutant, but in Figure 4, the data on increased TUNEL staining in thoracic ganglia and salivary glands is much more convincing in the xrRNA1' mutant, which was only mildly attenuated in the mosquito. This may in fact suggest that apoptosis is not the mediator for the reduced transmission that was observed. Additional experiments would provide more concrete evidence for the hypothetical role of apoptosis in reducing ZIKV xrRNA mutant transmission (e.g. using an apoptosis inhibitor to try to overcome the block in ZIKV transmission). In addition, including xrRNA1' mutant samples in the transcriptomic analysis may help elucidate what is happening or if other pathways/mechanisms are involved.

We have generated new *in vivo* data that supports our conclusion on antiapoptotic function of sfRNA (see also response to reviewer 1). We have demonstrated that treatment of mosquitoes with caspase inhibitor Z-VMD-FAK restored replication and transmission of both sfRNA-deficient mutants, but did not affect replication and transmission of WT virus (Fig 6 of the revised manuscript). We believe this now provides direct and convincing evidence for inhibition of apoptosis as the mechanism of sfRNA action.

We have now performed further statistical analysis of apoptosis rates in tissues of infected mosquitoes which showed no significant differences between xrRNA1 and xrRNA2 mutants in any of analysed tissues, with the exception of thoracic ganglia which showed border line significance (Fig S6, revised manuscript). We have also analysed apoptosis rates in mock-i.t.-injected mosquitoes (Fig 5 and Fig S6, revised manuscript) which showed increased apoptosis in thoracic ganglia of some mosquitoes, which could be the consequence of i.t. injection. However, we did not detect viral infection in this tissue after inoculation via blood feeding (data not shown) and no evidence was found in the literature for its role in flavivirus transmission. Therefore, we believe that apoptosis in thoracic ganglia is unlikely to influence ZIKV transmission or reflect the biological functions of sfRNA. Hence, we removed the panels with magnified view of thoracic ganglia from Fig 5 (Fig 4 of original manuscript).

Line-by-line comments:

Line 43: should be 'conducive to'

Corrected

Line 61: Suggestion: '...and the less abundant shorter sfRNA-2...'

Corrected

Line 67: The authors state: 'However, the exact role of sfRNA in arthropods is still unclear'. While the exact role is indeed unclear, the authors should highlight what is known and what potential roles for sfRNA have been proposed previously. For example, there are papers suggesting that Langkat virus and tick-borne encephalitis virus sfRNAs can inhibit RNA interference in ticks (Schnettler et al 2014), that dengue sfRNA may inhibit Toll signaling in salivary glands (Pompon et al. 2017), that WNV sfRNA1 is important for virus infection of the mosquito midgut and virus transmission (Goertz et al 2016) and that sfRNA has RNAi suppressor activity in mosquito cells (Schnettler et al. 2012; Moon et al. 2015). While some of these papers are cited later on, it would be useful to have a sentence or two here in the introduction providing some background on the various potential roles of sfRNAs in arthropods (or at least in mosquitoes). The introduction currently makes it sound as if this is the first study investigating this question.

We have amended the Introduction to incorporate previous finding as suggested by the reviewer (lines 61-64 of the revised manuscript). More extended description of findings that are relevant to our results is incorporated in the Discussion.

English: please check throughout for article usage. The definite article (and sometimes indefinite article) is missing at times (e.g. line 72: '... ZIKV replication in the insect host...' or as mentioned above in line 62). In line 100 should be '... contained a reversion...'

Corrected. The manuscript has been proof-read.

Figure 1 legend: more info could be provided, e.g. source of structure (crystal structure? model? How was it modelled?), time post transfection in B, add at least 'by Northern blot' or similar for C, and if error bars represent just standard deviation in D. Just a bit more detail would be useful for the reader.

We have now provided requested details in the figure legend. We have also extended the legends for all other figures to make sure they provide sufficient information required for understanding of the presented results.

Lines 96-102: The authors here describe an experiment using C6/36 cells to grow an sfRNA1+2 deficient virus, yet the measure of infectious virus produced is an immune-assay on Vero cells. The authors previously show that Vero cells do not support replication of this virus. So even if the xrRNA1'2 ZIKV mutant was able to replicate in C6/36 cells and generated infectious virus it would not be infectious on Vero cells (just C6/36 cells). The fact that a reversion occurred here does also not prove that the xrRNA1'2 ZIKV mutant virus cannot replicate in C6/36 – in fact, some replication must have occurred for the mutation to arise, or alternatively the cDNA prep was not clean and contained a small amount of xrRNA1' with intact xrRNA2. I would suggest using C6/36 cells as the read-out for infectious virus, as well as measuring extracellular viral RNA over time, and providing information on how many times this experiment was conducted. Currently, the conclusion that 'Therefore, generation of at least one sfRNA isoform is required for ZIKV viability in mosquito cells even with dysfunctional RNAi.' Is not warranted.

IPA on Vero cells was not the only readout used for virus detection in culture fluids of transfected C6/36 cells. We also tested culture fluids for the presence of extracellular viral RNA by RT-PCR, as described in the legend for Fig S1 of the original manuscript. However, when we sequenced the RT-PCR product we found that mutation reverted. Therefore, we don't see the need for measuring extracellular RNA in dynamics as it is no longer the expected mutant virus.

We would also like to clarify that we use the term "viral replication" in the meaning of "productive infection", i.e. formation of the infectious and secreted viruses during the infection process. In this regard, viral replication is not the same as replication of viral RNA. Viral RNA replication may not result in generation of infectious virus (productive infection) if other processes involved in viral replication (e.g virus assembly/secretion) are disrupted. To make our message clearer we have substituted "Viral replication" with "productive infection" wherever relevant in the revised manuscript.

We agree that initial replication of xrRNA1'2' mutant RNA could have occurred in transfected C6/36 cells, however, it didn't produce infectious viral particles as mutated genotype could not be detected in the culture fluids. This was repeated twice with the same result. This explanation is now added to the results (lines 102-108, revised manuscript).

The possibility that cDNA preps used in CPER assembly were not clean is highly unlikely as mutated cDNA fragments were sequenced prior to CPER assembly and mutations confirmed as stated in the "Methods" section.

Lines 114-121: this is repetitive and discussion. Also, the 'therefore' in line 114 is not a logical connection from the previous section where the generation two separate mutants for sfRNA 1 and 2 was already described.

Corrected

I would suggest to remove this section. In addition, my previous comments about replication of the double mutant in C6/36 cells negates a lot of the conclusions provided here without further evidence that the double mutant truly does not replicate in C6/36 cells.

We re-wrote this section of the manuscript to clarify the results (as described above) and believe that it is important to retain these results in the manuscript as they highlight the critical role of sfRNAs in infection.

Line 126: what were the titers used for bloodmeal and inoculation??

The titres are now presented in Fig S2A, D of the revised manuscript.

Figure 2: If the 'n=69' were moved into the middle of the graph, under the heading 'Blood feeding' it would be more obvious that it does not just refer to the wildtype bar, since it is currently situated right above that bar (same for the other panels). Presumably the 'n' was the same for each group (wt, xrRNA1', xrRNA2'). In the legend for panel A, it says 'inoculated mosquitoes' – please change to 'mosquitoes exposed to ZIKV by bloodmeal with a titer of ...' or similar, for clarity. Please also provided i.t. inoculation titers for B. In the last sentence an extra space should be removed ('tis sues' should be 'tissues').

Legends have been corrected. In the revised manuscript we specify sample sizes for each group for scatter plots in the figures and for bar graphs in the legends.

Line 131/132: This is a major drawback of this study. Why did the authors not try to obtain a more susceptible mosquito line? Are there specific reasons (containment, import etc?) why this wasn't possible?

As explained above, the reason was the strict Australian biosecurity regulations. However, as we were able to achieve higher infection rates in the local mosquitoes by using higher viral doses (Fig 2A-D in revised manuscript), we believe that this issue is now resolved.

Line 133ff: It should be noted somewhere in this section that sfRNA1 and sfRNA 3(?) were less abundant in the xrRNA2' mutant infected mosquitoes, which probably just correlates to lower viral RNA levels overall (since the blot is normalized to mosquito RNA not viral RNA). Clarifying this here would help the reader.

This is now stated and discussed in lines 189-193 and 366-373 of the revised manuscript respectively. There was no difference in viral RNA levels between the mutants as determined by qRT-PCR (Fig 2F in the revised manuscript).

Lines 148-149: the authors talk of 'dissemination to the legs and wings'. This is not a useful measure or description for an i.t. inoculation which initiates a disseminated infection without crossing the midgut barrier. It is reasonable to keep the data in, if desired although it does not add much information), but the word dissemination in vector competence studies refers to dissemination throughout the body after crossing the midgut barrier and is not appropriate here. Please re-word.

In the revised manuscript we no longer use the term "dissemination" to describe viral replication in legs and wings of i.t.-injected mosquitoes.

Lines 150-158: please check/correct English mistakes in this section for clarity.

Corrected

Line 173: the authors mention 'replication efficiency', however the viral titers from mosquito tissues were all obtained in Vero cells (presumably) where these mutants (especially xrRNA2', see Fig1B) are also attenuated. Other outputs such as viral RNA or a focus-forming assay using C6/36 cells could be useful to identify whether it is viral RNA replication, virus production or virus infectivity in Vero cells used to measure titers that's providing the (somewhat confusing) results. With the NS1 staining looking comparable, it is hard to imagine that viral RNA replication and infection of mosquito cells themselves are as significantly impacted as the titers insinuate.

The mutant viruses are not attenuated in Vero cells. See our detailed response on this to comment 3.

Line 206: here the authors mention that RNA content was comparable between wt and xrRNA2' mutant viruses suggesting that virus spread through the mosquito was probably not impacted, but not detected in the earlier experiments due to the 'virus titer on Vero cells' readout.

This appears to be related to the above comment. See our response above.

Figure 4: The x-axes are all different in Figure 4B, making it very hard to compare data.

Different tissues may have different viral loads and have different type of cells, likely with different activity of apoptotic pathways. The apoptosis levels should therefore not be compared between

different tissues but rather between different samples in each tissue. Hence, the x-axis scales were selected to enable the best resolution between the samples in a particular tissue.

In lines 225-228 the authors describe signs of apoptosis based on these data and I don't fully agree with the authors' conclusions (e.g. 'A limited number of mosquitoes had considerable signs of apoptosis in the head (Fig. 4B) with the only TUNEL-positive mosquitoes being those infected with mutant viruses' – there are wildtype data points that have some TUNEL staining, but the axis is very different to that for bodies, effectively reducing data resolution).

We reworded the text to make sure we accurately describe our data (lines 277-296, revised manuscript). We respectfully disagree with the reviewer that the scale of axes reduces data resolution. In fact, it is the opposite. Due to the presence of the individual outlier (see response to the comment below) the range of values for heads is highly skewed to the right and plotting data for other tissues on the same scale will compress them at the left and decrease data resolution.

Similarly, the authors state that 'Mutant viruses were also the only ones causing considerable apoptosis in salivary glands', when there is clearly at least one sample from wildtype virus that has significant TUNEL staining (more than any other mosquito). There are also xrRNA2' exposed mosquitoes without any virus in the salivary glands but significant TUNEL staining, which could indicate a confounding factor (or potentially TUNEL activation through extracellular factors).

We agree that some experimental groups contained single outliers, which is not uncommon for in vivo experiments. This is why we compare groups, and not individual mosquitoes. We have also performed additional analysis in which these samples with extreme values were identified as outliers by ROUT method with $Q=1\%$ (medium stringency). The contributing factors that cause outliers are usually hard to identify with confidence and it is possible that in this case some particular mosquitoes were injected more deeply or in a different, more sensitive, location than the majority of mosquitoes, which was causing more apoptosis. However, our analysis indicates that these events happen infrequently and have roughly equal occurrence among all groups including negative control. As these extreme values (i) have low occurrence, (ii) don't belong to the distributions we compare and (iii) are likely caused by the technical variations in the experimental procedure, we excluded them from the statistical analyses, although we still show them on the graphs (Fig 5, S6, revised manuscript).

In regard to TUNEL staining in areas without visible infection, it is well established that apoptosis can be induced by multiple factors and indeed could occur occasionally in uninfected tissues. In this study we are statistically testing the model in which apoptosis (dependent variable) is a function of sfRNA production (independent variable). We isolated one variable (sfRNA production) from other variables that may influence apoptosis by comparing the groups that are only different in sfRNA production (mutants vs WT). All other factors that induce apoptosis should act the same way across all groups and should not influence the outcome of the statistical tests. These factors will not be confounding our model as they influence dependent variable, but not the independent variable. In the revised manuscript we unknowledge that multiple factors can induce apoptosis and explain our statistical model (lines x-y, revised manuscript)

The data is in fact clearer for xrRNA1' mutants, which is counterintuitive considering these viruses are less attenuated compared to xrRNA2' mutants. It should also be noted that no uninfected control group was included. It would have been valuable to see TUNEL staining in uninfected mosquitoes for comparison.

Although TUNEL-staining of mosquitoes infected with xrRNA1' appears to look stronger, the additional statistical analysis that we have now performed in the revised manuscript showed that it is not significantly different from staining in the xrRNA2' group (Fig S6, revised manuscript). We also added TUNEL staining of the control group to Fig 5 of the revised manuscript. We did not observe considerable apoptosis in this group in all tissue except thoracic ganglia (see our detailed response on this to comment 5).

Line 240: please don't use 'ZIKV life cycle' or mosquito stage in this context. It is a transmission cycle if you want to give it a name as such. The sentence should just be restructured to avoid these terms.
Corrected

Line 264: The authors state: 'Thus, deficiency in sfRNAs appears to have a more profound effect on ZIKV infection than on WNV infection, although sfRNAs are clearly required for efficient replication and transmission in mosquitoes for both viruses'. I think it is worst repeating here why the authors think 'sfRNA appears to have a more profound effect on ZIKV...', i.e. because the authors used i.t. injection and still saw an effect. However, it is also hard to compare the results, since we don't know what would have happened after a successful infectious bloodmeal infection. Maybe the midgut aspect of the sfRNA is less pronounced in ZIKV, making the overall impact similar. I suggest elaborating a little bit here to be clear and so that only fair comparisons are made.

We now provide blood meal infection results and hence amended the discussion on this matter accordingly.

Methods:

Lines 311-313: please also provide the equivalent information/origin of RML-12 cells.

Provided (lines 544-545, revised manuscript)

Line 326: the authors haven't mentioned 'vero76 cells' what are these? What does the 76 stand for? Please add them in the 'cell culture' section.

Vero 76 is a cell line from ATCC that was derived from Vero cells, which has higher saturation density and was used in the experiments that require long incubation times. This is a commonly used cell line and information about it is available at ATCC web site. We have added this cell line to the "Methods" section and provided ATCC accession number.

Line 436: Please add 'Innisfail, Australia,' for clarity. The choice of using a nearly refractory mosquito line for these experiments seems a bit odd, although I understand if it was a matter of mosquito strain availability.

Added.

Line 440: Please provide information on virus titers in the blood meal. This may provide insight into the low infection rates observed and is important for the reader. Please also provide information on the ratios of blood and virus stock, as well as how virus stock concentrations were normalized (e.g. using culture media??).

We have provided virus titres (see Fig S2A in revised manuscript). We have also added information about the ratio of blood and virus stock to the methods.

Line 513: I assume the authors mean '60°C for 20 s' not minutes here, please correct.

Corrected.

Reviewers' Comments:

Reviewer #1:

Remarks to the Author:

This is a revision of a manuscript that I previously reviewed for this journal. The authors have convincingly addressed all my major comments. The authors chose not to follow my last suggestion, but they provide well-justified reasons not to do so.

In the revision, I encountered the following minor errors/typos:

- line 366, typo deficeint
- Legend to figure 6, panel indicators C and D should be B and C, respectively.
- Legend to Figure S2: Legend to panel B does not correspond to the figure panel. Legend to panel D is missing.

Reviewer #2:

Remarks to the Author:

The authors have addressed my concerns in detail and added relevant additional experiments/data. They have also further described/adjusted any aspects that were previously unclear to me and have increased clarity of the manuscript.

We would like to thank the reviewers for their time and consideration, and we are delighted that both reviewers were fully satisfied with our response to their comments and suggestions.

Reviewer 1 stated that

“The authors have convincingly addressed all my major comments. The authors chose not to follow my last suggestion, but they provide well-justified reasons not to do so”

Reviewer 1 also found the following errors and typos that we corrected in the revised manuscript:

-line 366, typo deficeint

-Legend to figure 6, panel indicators C and D should be B and C, respectively.

-Legend to Figure S2: Legend to panel B does not correspond to the figure panel. Legend to panel D is missing.

Reviewer #2 stated that

“The authors have addressed my concerns in detail and added relevant additional experiments/data. They have also further described/adjusted any aspects that were previously unclear to me and have increased clarity of the manuscript.”

Reviewer 2 hasn't raised any concerns.